# GNNDelete: A General Strategy for Unlearning in Graph Neural Networks

**Jiali Cheng[1]**    **George Dasoulas[*2]**    **Huan He[*2]**    **Chirag Agarwal[3]**    **Marinka Zitnik[2]**
[1]University of Massachusetts Lowell    [2]Harvard University    [3]Adobe
jiali_cheng@student.uml.edu    chiragagarwall12@gmail.com
{georgios_dasoulas,huan_he,marinka}@hms.harvard.edu
[*] indicates co-second authors

## Abstract

Graph unlearning, which involves deleting graph elements such as nodes, node labels, and relationships from a trained graph neural network (GNN) model, is crucial for real-world applications where data elements may become irrelevant, inaccurate, or privacy-sensitive. However, existing methods for graph unlearning either deteriorate model weights shared across all nodes or fail to effectively delete edges due to their strong dependence on local graph neighborhoods. To address these limitations, we introduce GNNDelete, a novel model-agnostic layer-wise operator that optimizes two critical properties, namely, Deleted Edge Consistency and Neighborhood Influence, for graph unlearning. Deleted Edge Consistency ensures that the influence of deleted elements is removed from both model weights and neighboring representations, while Neighborhood Influence guarantees that the remaining model knowledge is preserved after deletion. GNNDelete updates representations to delete nodes and edges from the model while retaining the rest of the learned knowledge. We conduct experiments on seven real-world graphs, showing that GNNDelete outperforms existing approaches by up to 38.8% (AUC) on edge, node, and node feature deletion tasks, and 32.2% on distinguishing deleted edges from non-deleted ones. Additionally, GNNDelete is efficient, taking 12.3x less time and 9.3x less space than retraining GNN from scratch on WordNet18.

## 1 Introduction

Graph neural networks (GNNs) are being increasingly used in a variety of real-world applications (Li et al., 2022a; Ying et al., 2019; Xu et al., 2022; 2019; Huang et al., 2021; Morselli Gysi et al., 2021; Hu et al., 2020), with the underlying graphs often evolving over time. Machine learning approaches typically involve offline training of a model on a complete training dataset, which is then used for inference without further updates. In contrast, online training methods allow for the model to be updated using new data points as they become available (Orabona, 2019; Nagabandi et al., 2019). However, neither offline nor online learning approaches can address the problem of data deletion (Cao & Yang, 2015b; Ginart et al., 2019), which involves removing the influence of a data point from a trained model without sacrificing model performance. When data needs to be deleted from a model, the model must be updated accordingly (Fu et al., 2022). In the face of evolving datasets and growing demands for privacy, GNNs must therefore not only generalize to new tasks and graphs but also be capable of effectively handling information deletion for graph elements from a trained model.

Despite the development of methods for machine unlearning, none of these approaches are applicable to GNNs due to fundamental differences arising from the dependencies between nodes connected by edges (which we show in this paper). Existing machine unlearning methods are unsuitable for data with underlying geometric and relational structure, as graph elements can exert a strong influence on other elements in their immediate vicinity. Furthermore, since the effectiveness of GNN models is based on the exchange of information across local graph neighborhoods, an adversarial agent can easily infer the presence of a data point from its neighbors if the impact of the data point on its local neighborhood is not limited. Given the wide range of GNN applications and the lack of graph unlearning methods, there is a pressing need to develop algorithms that enable GNN models to unlearn previously learned information. This would ensure that inaccurate, outdated, or

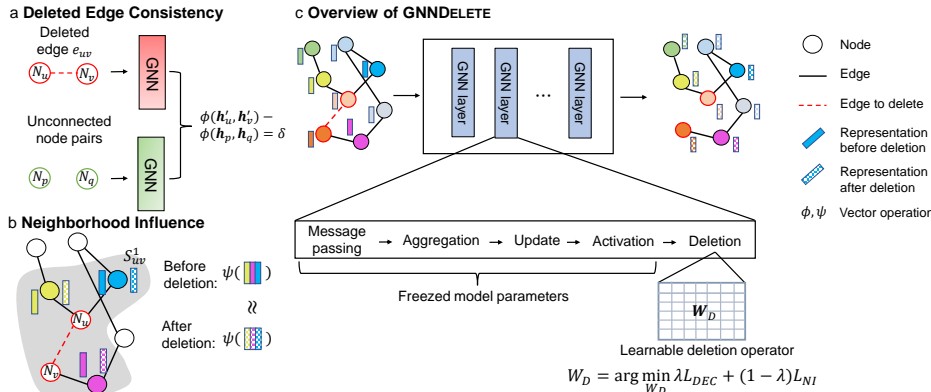

**Figure 1: a**. Illustration of Deleted Edge Consistency: It suggests that the predicted probability of deleted edges after unlearning should be random, such that it looks like the deleted data was not used for training before. **b**. Illustration of Neighborhood Influence: It implies that an appropriate unlearning should not change the representations of the local neighborhood (nodes in the subgraph, not nodes themselves ) to maintain the original causality. **c**. Overview of GNNDELETE: Given a trained GNN model and edge deletion request, GNNDELETE outputs unlearned representations efficiently by only learning a small deletion operator $W_D$. It also ensures representation quality by minimizing a loss function that satisfies the two key properties proposed above.

privacy-concerned graph elements are no longer used by the model, thereby preventing security concerns and performance degradation. In this paper, we take a step towards building an efficient and general-purpose graph unlearning method for GNNs.

Designing graph unlearning methods is a challenging task. Merely removing data is insufficient to comply with recent demands for increased data privacy because models trained on the original data may still contain information about removed features. A naive approach is to delete the data and retrain a model from scratch, but this can be prohibitively expensive, especially in large datasets. Recently, efforts have been made to achieve efficient unlearning based on exact unlearning (Brophy & Lowd, 2021; Sekhari et al., 2021; Hase et al., 2021; Ullah et al., 2021). The core idea is to retrain several independent models by dividing a dataset into separate shards and then aggregating their predictions during inference. Such methods guarantee the removal of all information associated with the deleted data. However, in the context of GNNs, dividing graphs destroys the structure of the input graph, leading to poor performance on node-, edge- and graph-level tasks. To address this issue, Chen et al. (2022b) uses a graph partitioning method to preserve graph structural information and aggregates predictions across individually retrained shards to produce predictions. However, this approach is still less efficient as the cost increases as the number of shards grows. In addition, choosing the optimal number of shards is still unresolved and may require extra hyperparameter tuning. Several approximation-based approaches (Guo et al., 2020; Ullah et al., 2021; He et al., 2021; Shibata et al., 2021) avoid retraining a model from scratch on data subsets. While these approaches have shown promise, Mitchell et al. (2022) demonstrated that these unlearning methods change the underlying predictive model in a way that can harm model performance.

**Present Work.** We introduce GNNDELETE[1], a general approach for graph unlearning that can delete nodes, node labels, and relationships from any trained GNN model. We formalize two essential properties that GNN deletion methods should satisfy: 1) *Deleted Edge Consistency:* predicted probabilities for deleted edges in the unlearned model should be similar to those for nonexistent edges. This property enforces GNNDELETE to unlearn information such that deleted edges appear as unconnected nodes. 2) *Neighborhood Influence:* we establish a connection between graph unlearning and Granger causality (Granger, 1969) to ensure that predictions in the local vicinity of the deletion maintain their original performance and are not affected by the deletion. However, existing graph unlearning methods do not consider this essential property, meaning they do not consider the influence of local connectivity, which can lead to sub-optimal deletion. To achieve both efficiency and scalability, GNNDELETE uses a layer-wise deletion operator to revise a trained GNN model. When receiving deletion requests, GNNDELETE freezes the model weights and learns additional small weight matrices that are shared across nodes in the graph. Unlike methods that attempt to

---

[1]Code and datasets for GNNDELETE can be found at https://github.com/mims-harvard/GNNDelete.

retrain several small models from scratch or directly update model weights, which can be inefficient and suboptimal, GNNDELETE learns small matrices for inference without changing GNN model weights. To optimize GNNDELETE, we specify a novel objective function that satisfies Deleted Edge Consistency and Neighborhood Influence and achieves strong deletion performance.

**Our Contributions.** We present our contributions as follows: ① We formalize the problem of graph unlearning and define two key properties, *Deleted Edge Consistency* and *Neighborhood Influence*, for effective unlearning on graph data. ② We propose GNNDELETE, a GNN data deletion approach that achieves more efficient unlearning than existing methods without sacrificing predictive performance. GNNDELETE is model-agnostic and can be applied to various GNN architectures and training methodologies. ③ The difference between node representations returned the baseline GNN model and those revised by GNNDELETE is theoretically bounded, ensuring the strong performance of GNNDELETE. ④ We demonstrate the flexibility of GNNDELETE through empirical evaluations on both link and node deletion tasks. Results show that GNNDELETE achieves effective unlearning with 12.3x less time and 9.3x less computation than retraining from scratch.

## 2 RELATED WORK

**Machine Unlearning.** We organize machine unlearning research into four categories: 1) *Retraining:* Retraining models from scratch to unlearn is a simple yet often inefficient approach, despite recent efforts to develop efficient data partitioning and retraining methods (Bourtoule et al., 2021; Wu et al., 2020; Liu et al., 2022b; Cao & Yang, 2015a; Golatkar et al., 2020; Izzo et al., 2021). However, partitioning graphs can be challenging because the graph structure and learned node representations of the partitioned graphs may be significantly different from the original graph. Furthermore, such methods may not scale well to large datasets. 2) *Output modification:* Methods such as UNSIR (Tarun et al., 2021) directly modify model outputs to reduce computational overhead. UNSIR first learns an error matrix that destroys the output and then trains the destroyed model for two epochs with clean data to repair the outputs. However, for graphs, the error destroys outputs for all edges, and the training after that falls back to retraining the whole model. 3) *Logit manipulation:* Other methods achieve unlearning by manipulating the model logits (Izzo et al., 2021; Baumhauer et al., 2022), but these methods only apply to linear or logit-based models. 4) *Weight modification:* Unlearning via weight modification is achieved by running an optimization algorithm. For example, Ullah et al. (2021) proposed an unlearning method based on noisy stochastic gradient descent, while Guo et al. (2020) achieves certified removal based on Newton updates. Other optimization methods that modify weights include Thudi et al. (2022a) and Neel et al. (2021). Recent unlearning methods perturb gradients (Ma et al., 2022) or model weights (Chen et al., 2021a). However, weight modification approaches lack unique features for graphs and incur computation overheads, such as calculating the inverse Hessian. In Appendix A, we provide details on unlearning methods for other models.

**Graph Unlearning.** We present an overview of the current state of the art in graph unlearning research. GraphEraser (Chen et al., 2022b) attempts to address the graph unlearning problem by utilizing graph partitioning and efficient retraining. They use a clustering algorithm to divide a graph into shards based on both node features and structural information. A learnable aggregator is optimized to combine the predictions from sharded models. However, the limitations of GraphEraser (Chen et al., 2022b) are that it supports only node deletion. GraphEditor (Cong & Mahdavi, 2023) provides a closed-form solution for linear GNNs to guarantee information deletion, including node deletion, edge deletion, and node feature update. Additional fine-tuning can improve predictive performance. However, GraphEditor is only applicable to linear structures, which is the case for most unlearning algorithms, not only those designed for graph-structured data. As a result, it is not possible to use existing non-linear GNNs or knowledge graphs with GraphEditor, and it struggles to process larger deletion requests. Recently, Chien et al. (2022) proposed the first framework for certified graph unlearning of GNNs. Their approach provides theoretical guarantees for approximate graph unlearning. However, the framework is currently limited to certain GNN architectures and requires further development to become a more practical solution for the broader range of GNNs. For more details on related work, we refer the reader to Appendix A.

**Connection with Adversarial Attacks and Defense for GNNs.** To determine whether a data point has been used to train a model, the success of a membership inference (MI) attack can be a suitable measure for the quality of unlearning (Yeom et al., 2019; Sablayrolles et al., 2019). Defending

against MI attacks is also a challenge that we care about when building unlearning models. Thudi et al. (2022c) proposed using a novel privacy amplification scheme based on a new tighter bound and subsampling strategy. Olatunji et al. (2021) showed that all GNN models are vulnerable to MI attacks and proposed two defense mechanisms based on output perturbation and query neighborhood perturbation. Liu et al. (2022a) treated the data to be unlearned as backdoored data. While defense strategies against MI attacks can provide valuable insights for evaluating unlearning, it is important to note that they serve a different purpose than unlearning itself.

## 3 PRELIMINARIES

Let $G = (\mathcal{V}, \mathcal{E}, \boldsymbol{X})$ be an attributed graph with $n = |\mathcal{V}|$ nodes, set of edges $\mathcal{E}$, and $n_f$-dimensional node features $\boldsymbol{X} = \{\boldsymbol{x}_0, \ldots, \boldsymbol{x}_{n-1}\}$ where $\boldsymbol{x}_i \in \mathbb{R}^{n_f}$. We use $\boldsymbol{A}$ to denote the adjacency matrix of $G$ and $\deg_G : \mathcal{V} \to \mathbb{N}$ to denote the degree distribution of graph $G$. Further, we use $\mathcal{S}_{uv}^k = (\mathcal{V}_{uv}^k, \mathcal{E}_{uv}^k, \boldsymbol{X}_{uv}^k)$ to represent a $k$-hop enclosing subgraph around nodes $u$ and $v$.

**Graph Neural Networks (GNNs).** A GNN layer $g$ can be expressed as a series of transformation functions: $g(G) = (\text{UPD} \circ \text{AGG} \circ \text{MSG})(G)$ that takes $G$ as input and produces $n$ $d$-dimensional node representations $\boldsymbol{h}_u$ for $u \in \mathcal{V}$ (Figure 1). Within layer $l$, MSG specifies neural messages that are exchanged between nodes $u$ and $v$ following edges in $\boldsymbol{A}_{uv}$ by calculating $\boldsymbol{p}_{uv}^l = \text{MSG}(\boldsymbol{h}_u^{l-1}, \boldsymbol{h}_v^{l-1}, \boldsymbol{A}_{uv})$. The AGG defines how every node $u$ combines neural messages from its neighbors $\mathcal{N}_u$ and computes the aggregated message $\boldsymbol{P}_u^l = \text{AGG}(\{\boldsymbol{p}_{uv}^l | v \in \mathcal{N}_u\})$. Finally, UPD defines how the aggregated messages $\boldsymbol{P}_u^l$ and hidden node states from the previous layer are combined to produce $\boldsymbol{h}_u^l$, i.e., final outputs of $l$-th layer $\boldsymbol{h}_u^l = \text{UPD}(\boldsymbol{P}_u^l, \boldsymbol{h}_u^{l-1})$. The output of the last GNN layer is the final node representation, $\boldsymbol{z}_u = \boldsymbol{h}_u^L$, where $L$ is the number of GNN layers in the model.

**Unlearning for GNNs.** Let $\mathcal{E}_d \subseteq \mathcal{E}$ denote the set of edges to be deleted and $\mathcal{E}_r = \mathcal{E} \backslash \mathcal{E}_d$ be the remaining edges after the deletion of $\mathcal{E}_d$ from $G$. We use $G_r = (\mathcal{V}_r, \mathcal{E}_r, \boldsymbol{X}_r)$ to represent the resulting graph after deleting edge $\mathcal{E}_d$. Here, $\mathcal{V}_r = \{u \in V | \deg_{G_r}(u) > 0\}$ denotes the set of nodes that are still connected to nodes in $G_r$, and $\boldsymbol{X}_r$ denotes the corresponding node attributes. Although the above notations are specific to edge deletion, GNNDELETE can also be applied to node deletion by removing all edges incident to the node that needs to be deleted from the model.

To unlearn an edge $e_{uv}$, the model must erase all the information and influence associated with $e_{uv}$ as if it was never seen during the training while minimizing the change in downstream performance. To this end, we need to modify both the predicted probability of $e_{uv}$ and remove its information from its local neighborhood. Therefore, post-processing and logit manipulation are ineffective for deleting edges from a GNN model because these strategies do not affect the rest of the graph. We denote a classification layer $f$ that takes node representations $\boldsymbol{h}_u^L$ and $\boldsymbol{h}_v^L$ as input and outputs the prediction probability for edge $e_{uv}$. Given $L$ layers and $f$, a GNN model $m : G \to \mathbb{R}^{|\mathcal{E}|}$ can be expressed as $m(G) = (f \circ g^L \cdots \circ g^1)(G)$, where $g^i$ is the $i^{th}$ GNN layer. The unlearned model $m' : G \to \mathbb{R}^{|\mathcal{E}_r|}$ can be written as $m'(G_r) = (f \circ g'^L \cdots \circ g'^1)(G_r)$, which are $L$ stacked unlearned GNN layers $g'^i$ operating on the graph $G_r$.

## 4 GNNDELETE: A GENERAL STRATEGY FOR GRAPH UNLEARNING

To ensure effective edge deletion from graphs, the GNN model should ignore edges in $\mathcal{E}_d$ and not be able to recognize whether a deleted edge $e_{uv} \in \mathcal{E}_d$ is part of the graph. Furthermore, the model should ignore any influence that a deleted edge has in its neighborhood. To this end, we introduce two properties for effective graph unlearning and a layer-wise deletion operator that implements the properties and can be used with any GNN to process deletions in $\mathcal{E}_d$.

**Problem Formulation (Graph Unlearning).** *Given a graph $G = (\mathcal{V}, \mathcal{E}, \boldsymbol{X})$ and a fully trained GNN model $m(G)$, we aim to unlearn every edge $e_{uv} \in \mathcal{E}_d$ from $m(G)$, where $\mathcal{E}_d$ is a set of edges to be deleted. The goal is to obtain an unlearned model $m'(G)$ that is close to the model output that would have been obtained had the edges in $\mathcal{E}_d$ been omitted from training. To achieve this, we require that the following properties hold:*

- *Deleted Edge Consistency: If $e_{uv} \in \mathcal{E}_d$, then $m'(G)$ should output a prediction that is independent of the existence of the edge $e_{uv}$, i.e., the deletion of $e_{uv}$ should not have any influence on the predicted output.*

- *Neighborhood Influence: If $e_{uv} \notin \mathcal{E}_d$, then $m'(G)$ should output a prediction that is close to $m(G)$, i.e., the deletion of edges in $\mathcal{E}_d$ should not have any significant impact on predictions in the rest of the graph.*

## 4.1 REQUIRED PROPERTIES FOR SUCCESSFUL DELETION ON GRAPHS

Deleting information from a graph is not a trivial task because the representations of nodes and edges are dependent on the combined neighborhood representations. The following two properties show intuitive assumptions over the deletion operator for effective unlearning in GNNs:

**1) Deleted Edge Consistency.** The predicted probability from the unlearned model $m'$ for an edge $e_{uv}$ should be such that it is hard to determine whether it is a true edge or not. The unlearned GNN layer $g'^l$ should not be aware of the edge existence. Formally, we define the following property:

**Definition 1** (Deleted Edge Consistency). *Let $e_{uv}$ denote an edge to be deleted, $g^l$ be the l-th layer in a GNN with output node representation vectors $\boldsymbol{h}_u^l$, and the unlearned GNN layer $g'^l$ with $\boldsymbol{h}_u'^l$. The unlearned layer $g'^l$ satisfies the Deleted Edge Consistency property if it minimizes the difference between node-pair representations $\phi(\boldsymbol{h}_u', \boldsymbol{h}_v')$, and $\phi(\boldsymbol{h}_p, \boldsymbol{h}_q)$ of two randomly chosen nodes $p, q \in \mathcal{V}$:*

$$\underset{p,q \in_R \mathcal{V}}{\mathbb{E}} [\phi(\boldsymbol{h}_u'^l, \boldsymbol{h}_v'^l) - \phi(\boldsymbol{h}_p, \boldsymbol{h}_q)] = \delta, \tag{1}$$

where $\phi$ is a readout function (e.g., dot product, concatenation) that combines node representations $\boldsymbol{h}_u'^l$ and $\boldsymbol{h}_v'^l$, $\in_R$ denotes a random choice from $\mathcal{V}$, and $\delta$ is an infinitesimal constant.

**2) Neighborhood Influence.** While the notion of causality has been used in explainable machine learning, to the best of our knowledge, we propose the first effort of modifying a knowledge graph using a causal perspective. Formally, removing edge $e_{uv}$ from the graph requires unlearning the influence of $e_{uv}$ from the subgraphs of both nodes $u$ and $v$. In this work, we propose the *Neighborhood Influence* property which leverages the notion of Granger causality (Granger, 1969; Bressler & Seth, 2011) and declares a causal relationship $\psi(\{\boldsymbol{h}_u | u \in \mathcal{S}_{uv}\}) \rightarrow e_{uv}$ between variables $\psi(\{\boldsymbol{h}_u | u \in \mathcal{S}_{uv}\})$ and $e_{uv}$ if we are better able to predict edge $e_{uv}$ using all available node representations in $\mathcal{S}_{uv}$ than if the information apart from $\psi(\{\boldsymbol{h}_u | u \in \mathcal{S}_{uv}\})$ had been used. Here, $\psi(\cdot)$ is an operator that combines the node representations in subgraph $\mathcal{S}_{uv}$. In the context of graph unlearning, if the absence of node representations decreases the prediction confidence of $e_{uv}$, then there is a causal relationship between the node representation and the $e_{uv}$ prediction.

Here, we characterize the notion of deletion by extending Granger causality to local subgraph causality, i.e., an edge $e_{uv}$ dependent on the subgraphs associated with both nodes $u$ and $v$. In particular, removing $e_{uv}$ should not affect the predictions of $\mathcal{S}_{uv}$ yielding the following property:

**Definition 2** (Neighborhood Influence). *Let $e_{uv} \in \mathcal{E}_d$ denote an edge in G to be deleted, $g^l$ be the l-th layer in a GNN with output node representation vectors $\boldsymbol{h}_u^l$, and the unlearned GNN layer $g'^l$ with $\boldsymbol{h}_u'^l$. The unlearned layer $g'^l$ satisfies the Neighborhood Influence property if it minimizes the difference of all node-subset representations $\psi(\{\boldsymbol{h}_w^l | w \in \mathcal{S}_{uv}\})$ comprising $e_{uv}$ with their corresponding node-subset representations $\psi(\{\boldsymbol{h}_w'^l | w \in \mathcal{S}_{uv/e_{uv}}\})$ where $e_{uv}$ is deleted, i.e.,*

$$\psi(\{\boldsymbol{h}_w^l | w \in \mathcal{S}_{uv}\}) - \psi(\{\boldsymbol{h}_w'^l | w \in \mathcal{S}_{uv/e_{uv}}\})] = \delta, \tag{2}$$

where $\psi$ is an operator that combines the elements of $S_{uv}$ (e.g., concatenation, summation), $\mathcal{S}_{uv/e_{uv}}$ represent the subgraph excluding the information from $e_{ij}$, and $\delta$ is an infinitesimal constant.

## 4.2 LAYER-WISE DELETION OPERATOR

To achieve effective deletion of an edge $e_{uv}$ from a graph $G$, it is important to eliminate signals with minor contributions to predicting the edges and develop mechanisms that can tune or perturb any source of node or edge information that aids in the prediction of $e_{uv}$. Perturbing weights or other hyperparameters of the GNN model can affect decisions for multiple nodes and edges in $G$ due to information propagation through the local neighborhood of each node. In order to allow for the deletion of specific nodes and edges, we introduce a model-agnostic deletion operator DEL that can be applied to any GNN layer.

**Deletion Operator.** Following the notations of Section 3, for the l-th GNN layer $g^l(G)$ with output dimension $d^l$, we define an extended GNN layer with unlearning capability as $(\text{DEL}^l \circ g^l)(G)$ with

the same output dimension $d^l$. Given an edge $e_{uv}$ that is to be removed, DEL is applied to the node representations and is defined as:

$$\text{DEL}^l = \begin{cases} \phi & \text{if } w \in S_{uv}^l \\ \mathbb{1} & \text{otherwise} \end{cases}, \tag{3}$$

where $\mathbb{1}$ is the identity function, and $\phi : \mathbb{R}^{n \times d^l} \to \mathbb{R}^{n \times d^l}$ can be any differentiable function that takes as input the output node representations of $g$. In this work, $\phi$ is considered as an MLP with weight parameters $\boldsymbol{W}_D^l$. Similarly to other GNN operators, the weights $\boldsymbol{W}_D^l$ of our DEL operator are shared across all nodes to achieve efficiency and scalability.

**Local Update.** Defining an operator that acts only in the local neighborhood $\mathcal{S}_{uv}$ enables targeted unlearning, keeping the previously learned knowledge intact as much as possible. If node $u$ is within the local neighborhood of $e_{uv}$, DEL is activated. For other nodes, DEL remains deactivated and does not affect the hidden states of the nodes. This ensures that the model will not forget the knowledge it has gained before during training and the predictive performance on $\mathcal{E} \backslash S_{uv}$ will not drop.

By applying the deletion operator DEL to every GNN layer, we expect the final representations to reflect the unlearned information in the downstream task. Next, we show a theoretical observation over the unlearned node representations that indicates a stable behavior of the deletion operator:

**Theorem 1.** *(Bounding edge prediction using initial model $m$ and unlearned model $m'$) Let $e_{uv}$ be an edge to be removed, $\boldsymbol{W}_D^L$ be the weight matrix of the deletion operator $\text{DEL}^L$, and normalized Lipschitz activation function $\sigma(\cdot)$. Then, the norm difference between the dot product of the final node representations from the initial model $\boldsymbol{z}_u, \boldsymbol{z}_v$ and from the unlearned one $\boldsymbol{z}_u', \boldsymbol{z}_v'$ is bounded by:*

$$\langle \boldsymbol{z}_u, \boldsymbol{z}_v \rangle - \langle \boldsymbol{z}_u', \boldsymbol{z}_v' \rangle \geq -\frac{1 + \|\boldsymbol{W}_D^L\|^2}{2} \|\boldsymbol{z}_u - \boldsymbol{z}_v\|^2, \tag{4}$$

*where $\boldsymbol{W}_D^L$ denotes the weight matrix of the deletion operator for the $l$-th GNN layer.*

The proof is in Appendix B. By Theorem 1, $\langle \boldsymbol{z}_u', \boldsymbol{z}_v' \rangle$, and consequently the prediction probability for edge $e_{uv}$ from the unlearned model cannot be dissimilar from the baseline. Nevertheless, $\text{DEL}^l$ is a layer-wise operator, which provides stable node embeddings as compared to the initial ones.

## 4.3 MODEL UNLEARNING

Moving from a layer-wise operator to the whole GNN model, our method GNNDELETE applies DEL to every GNN layer, leading to a total number of trainable parameters $\sum_l (d^l)^2$. As the number of trainable parameters in GNNDELETE is independent of the size of the graph, it is compact and scalable to larger graphs and the number of deletion requests. Considering the properties defined in Section 4.1, we design two loss functions and compute them in a layer-wise manner. Specifically, for the $l$-th GNN layer we first compute the *Deleted Edge Consistency* loss:

$$\mathcal{L}_{\text{DEC}}^l = \mathcal{L}(\{[\boldsymbol{h}_u'^l; \boldsymbol{h}_v'^l] | e_{uv} \in \mathcal{E}_d\}, \{[\boldsymbol{h}_u^l; \boldsymbol{h}_v^l] | u, v \in_R \mathcal{V}\}), \tag{5}$$

and the *Neighborhood Influence* loss:

$$\mathcal{L}_{\text{NI}}^l = \mathcal{L}(\|_w \{\boldsymbol{h}_w'^l | w \in \mathcal{S}_{uv}^l / e_{uv}\}, \|_w \{\boldsymbol{h}_w^l | w \in \mathcal{S}_{uv}^l\}), \tag{6}$$

where $[\boldsymbol{h}_u'^l; \boldsymbol{h}_v'^l]$ denotes the concatenation of two vectors, and $\|$ denotes the concatenation of multiple vectors. Note that according to Equations 1 and 2, we choose the functions $\phi, \psi$ to be the concatenation operators. During the backward pass, the deletion operator at the $l$-th GNN layer is only optimized based on the weighted total loss at the $l$-th layer, i.e.

$$\boldsymbol{W}_D^{l^*} = \arg\min_{\boldsymbol{W}_D^l} \mathcal{L}^l = \arg\min_{\boldsymbol{W}_D^l} \lambda \mathcal{L}_{\text{DEC}}^l + (1 - \lambda) \mathcal{L}_{\text{NI}}^l, \tag{7}$$

where $\lambda \in [0, 1]$ is a regularization coefficient that balances the trade-off between the two properties, $\mathcal{L}$ refers to the distance function. We use Mean Squared Error (MSE) throughout the experiments.

**Broad Applicability of GNNDELETE.** GNNDELETE treats node representations in a model-agnostic manner, allowing us to consider graph unlearning in models beyond GNNs. Graph transformers (Ying et al., 2021; Rampášek et al., 2022) have been proposed recently as an extension

of the Transformer architecture (Vaswani et al., 2017) for learning representations on graphs. The DEL operator can also be applied after the computation of the node representations in such models. For example, the $\text{MPNN}_e^l(\boldsymbol{X}^l, \boldsymbol{E}^l, \boldsymbol{A})$ layer in GraphGPS (Rampášek et al., 2022, Equation 2) can be replaced with the unlearned version ($\text{DEL} \circ \text{MPNN}_e^l$). Similarly, in the Graphormer layer, DEL operator can be applied after the multi-head attention MHA layer (Rampášek et al., 2022, Equation 8).

## 5 EXPERIMENTS

We proceed with the empirical evaluation of GNNDELETE. We examine the following questions: **Q1**) How does GNNDELETE perform compared to existing state-of-the-art unlearning methods? **Q2**) Can GNNDELETE support various unlearning tasks including node, node label, and edge deletion? **Q3**) How does the interplay between Deleted Edge Consistency and Neighborhood Influence property affect deletion performance? Appendix C.1 provides a detailed definition of performance metrics.

### 5.1 EXPERIMENTAL SETUP

**Datasets.** We evaluate GNNDELETE on several widely-used graphs at various scales. We use 5 homogeneous graphs: Cora (Bojchevski & Günnemann, 2018), PubMed (Bojchevski & Günnemann, 2018), DBLP (Bojchevski & Günnemann, 2018), CS (Bojchevski & Günnemann, 2018), OGB-Collab (Hu et al., 2020), and 2 heterogeneous graphs: OGB-BioKG (Hu et al., 2020), and WordNet18RR (Dettmers et al., 2018). Table 4 includes details on graph datasets.

**GNNs and Baselines.** We test with four GNN architectures and two graph types to show the flexibility of our GNNDELETE operator. In particular, we test on GCN (Kipf & Welling, 2017), GAT (Veličković et al., 2018), and GIN (Xu et al., 2019) for homogeneous graphs, and R-GCN (Schlichtkrull et al., 2018) and R-GAT (Chen et al., 2021b) for heterogeneous graphs. We consider four baseline methods: i) GRAPHEDITOR (Cong & Mahdavi, 2023), a method that finetunes on a closed-form solution of linear GNN models; ii) CERTUNLEARN (Chien et al., 2022), a certified unlearning approach based on linear GNNs; iii) GRAPHERASER (Chen et al., 2022b), a re-training-based machine unlearning method for graphs; iv) GRADASCENT, which performs gradient ascent on $\mathcal{E}_d$ with cross-entropy loss, and v) DESCENT-TO-DELETE (Neel et al., 2021), a general machine unlearning method.

**Unlearning Tasks and Downstream Tasks.** Requests for graph unlearning can be broadly classified into three categories: 1) edge deletion, which involves removing a set of edges $\mathcal{E}_d$ from the training graph, 2) node deletion, which involves removing a set of nodes $\mathcal{N}_d$ from the training graph, and 3) node feature unlearning, which involves removing the node feature $X_d$ from the nodes $\mathcal{N}_d$. Deletion of information can have a significant impact on several downstream tasks. Therefore, we evaluate the effects of graph unlearning on three different downstream tasks, namely, link prediction, node classification, and graph classification.

**Setup.** We evaluate the effectiveness of GNNDELETE on edge deletion tasks and also demonstrate its ability to handle node deletion and node feature unlearning tasks. We perform experiments on two settings: i) an easier setting where we delete information far away from test set $\mathcal{E}_t$ in the graph, and ii) a harder setting where we delete information proximal to test set $\mathcal{E}_t$ in the graph. To perform edge deletion tasks, we delete a varying proportion of edges in $\mathcal{E}_d$ between [0.5%-5.0%] of the total edges, with a step size of 0.5%. For larger datasets such as OGB (Hu et al., 2020), we limit the maximum deletion ratio to 2.5%. We report the average and standard error of the unlearning performance across five independent runs. We use AUROC to evaluate the performance of GNNDELETE for link prediction tasks, as well as Membership Inference (MI) (Thudi et al., 2022a) for node deletion. Performance metrics are described in Appendix C.1. Additionally, we consider two sampling strategies for $\mathcal{E}_d$ (Appendix C.2).

### 5.2 **Q1**: RESULTS – COMPARISON TO EXISTING UNLEARNING STRATEGIES

We compare GNNDELETE to the baseline unlearning techniques and present the results in Tables 1. Across four GNN architectures, we find that GNNDELETE achieves the best performance on the test edge set $\mathcal{E}_t$, outperforming GRAPHEDITOR, CERTUNLEARN and GRAPHERASER by 13.9%, 19.7% and 38.8%. Further, we observe that GNNDELETE achieves the highest AUROC on $\mathcal{E}_d$,

**Table 1: Unlearning task: 2.5% edge deletion. Evaluation: link prediction. Dataset: DBLP and Word-Net18**. Reported is AUROC (↑) performance with the best marked in **bold** and the second best underlined. GNNDELETE retains the predictive power of GNNs on $\mathcal{E}_t$ while improving performance on $\mathcal{E}_d$. GRAPHEDITOR and CERTUNLEARN support unlearning for linear GNNs on homogeneous graphs (i.e., N/A). Table 7 shows results on other downstream tasks and Tables 15-22 show results on other datasets and ratios of deleted edges.

| Model | GCN | | GAT | | R-GCN | | R-GAT | |
|---|---|---|---|---|---|---|---|---|
| | $\mathcal{E}_t$ | $\mathcal{E}_d$ | $\mathcal{E}_t$ | $\mathcal{E}_d$ | $\mathcal{E}_t$ | $\mathcal{E}_d$ | $\mathcal{E}_t$ | $\mathcal{E}_d$ |
| RETRAIN | 0.964 ±0.003 | 0.506 ±0.013 | 0.956 ±0.002 | 0.525 ±0.012 | 0.800 ±0.005 | 0.580 ±0.006 | 0.891 ±0.005 | 0.783 ±0.009 |
| GRADASCENT | 0.555 ±0.066 | 0.594 ±0.063 | 0.501 ±0.020 | 0.592 ±0.017 | 0.490 ±0.001 | 0.502 ±0.002 | 0.490 ±0.001 | 0.492 ±0.003 |
| D2D | 0.500 ±0.000 | 0.500 ±0.000 | 0.500 ±0.000 | 0.500 ±0.000 | 0.500 ±0.000 | 0.500 ±0.000 | 0.500 ±0.000 | 0.500 ±0.000 |
| GRAPHERASER | 0.527 ±0.002 | 0.500 ±0.000 | 0.538 ±0.013 | 0.500 ±0.000 | 0.512 ±0.003 | 0.500 ±0.000 | 0.545 ±0.015 | 0.500 ±0.000 |
| GRAPHEDITOR | 0.776 ±0.025 | 0.432 ±0.009 | - | - | N/A | N/A | N/A | N/A |
| CERTUNLEARN | 0.718 ±0.032 | 0.475 ±0.011 | - | - | N/A | N/A | N/A | N/A |
| GNNDELETE | **0.934 ±0.002** | **0.748 ±0.006** | **0.914 ±0.007** | **0.774 ±0.015** | **0.751 ±0.006** | **0.845 ±0.007** | **0.893 ±0.002** | **0.786 ±0.004** |

**Table 2: Unlearning task: 2.5% edge deletion. Evaluation: MI attack. Dataset: DBLP and WordNet18**. Reported is the MI attack ratio (↑), with the best performance marked in **bold** and the second best underlined. GRAPHEDITOR and CERTUNLEARN support unlearning for linear GNNs on homogeneous graphs (i.e., N/A).

| | DBLP | | | WordNet18 | |
|---|---|---|---|---|---|
| Model | GCN | GAT | GIN | R-GCN | R-GAT |
| RETRAIN | 1.255 ±0.207 | 1.223 ±0.151 | 1.200 ±0.177 | 1.250 ±0.091 | 1.215 ±0.125 |
| GRADASCENT | 1.180 ±0.061 | 1.112 ±0.109 | 1.123 ±0.103 | 1.169 ±0.066 | 1.112 ±0.106 |
| D2D | 1.264 ±0.000 | 1.264 ±0.000 | **1.264 ±0.000** | **1.268 ±0.000** | 1.268 ±0.000 |
| GRAPHERASER | 1.101 ±0.032 | 1.182 ±0.104 | 1.071 ±0.113 | 1.199 ±0.048 | 1.173 ±0.104 |
| GRAPHEDITOR | 1.189 ±0.193 | 1.189 ±0.193 | 1.189 ±0.193 | N/A | N/A |
| CERTUNLEARN | 1.103 ±0.087 | 1.103 ±0.087 | 1.103 ±0.087 | N/A | N/A |
| GNNDELETE | **1.266 ±0.106** | **1.338 ±0.122** | 1.254 ±0.159 | 1.264 ±0.143 | **1.280 ±0.144** |

outperforming GRAPHEDITOR, CERTUNLEARN and GRAPHERASER by 32.2%, 27.9% and 25.4%. GNNDELETE even outperforms RETRAIN-FROM-SCRATCH by 21.7% under this setting, proving its capability of effectively unlearning the deleted edges. Interestingly, none of the existing baseline methods have comparable performance to GNNDELETE on these performance metrics, including GRAPHERASER, which ignores the global connectivity pattern and overfit to specific shards, as well as GRAPHEDITOR and CERTUNLEARN, whose choice of linear architecture strongly limits the power of the GNN unlearning. Our results demonstrate that baselines like DESCENT-TO-DELETE and GRADASCENT lose almost their predictive prowess in making meaningful predictions and distinguishing deleted edges because the weight updates are independent of the unlearning task and affect all the nodes, including nodes associated with $\mathcal{E}_t$. In addition, CERTUNLEARN and GRAPHEDITOR are not applicable due to their linear architectures. Please refer to the Appendix for results on Cora (Tables 13-14), PubMed (Tables 15-16), DBLP (Tables 17-18), OGB-Collab (Tables 19-20), and WordNet18 (Tables 21-22) using a deletion ratio of 0.5%, 2.5%, and 5%.

Results in Table 2 show the Membership Inference (MI) performance of baselines and GNNDELETE for the DBLP and Wordnet18 using a deletion ratio of 2.5%. It shows that GNNDELETE outperforms baselines for most GNN models, highlighting its effectiveness in hiding deleted data. Across five GNN architectures, we find that GNNDELETE improves on the MI ratio score of all baselines: GRAPHEDITOR (+0.083), CERTUNLEARN (+0.169) GRAPHERASER (+0.154), RETRAIN-FROM-SCRATCH (+0.047), GRADASCENT (+0.134), and DESCENT-TO-DELETE (+0.086).

## 5.3 Q2: RESULTS – OTHER UNLEARNING TASKS AND GNNDELETE'S EFFICIENCY

**Node Deletion.** We examine the flexibility of GNNDELETE to handle node deletion. We delete 100 nodes and their associated edges from the training data and evaluate the performance of the unlearning method on node classification and Membership Inference attacks. Results in Table 8 show that GNNDELETE outperforms baselines on node classification while deleting nodes. GNNDELETE outperforms GRAPHEDITOR and CERTUNLEARN by 4.7% and 4.0% in accuracy, respectively. It is also 0.139 and 0.267 better than GRAPHEDITOR and CERTUNLEARN in terms of membership inference attacks. Tables 9 and 10 show results for node feature unlearning and sequential unlearning.

**Time and Space Efficiency.** We demonstrate that GNNDELETE is time-efficient as compared to most unlearning baselines. For all methods, we use a 2-layer GCN/R-GCN architecture with a trainable entity and relation embeddings with 128, 64, and 32 hidden dimensions trained on three datasets (PubMed, CS, and OGB-Collab). We present the results of wall-clock time vs. graph size in Figure 2 and observe that GNNDELETE consistently takes less time than existing graph unlearning methods. In particular, GNNDELETE is **12.3×** faster than RETRAIN-FROM-SCRATCH on WordNet. For smaller graphs like DBLP, GNNDELETE takes 185 seconds less (18.5% faster) than the pre-training stage of GRAPHEDITOR. Despite taking lower time, the predictive performances of DESCENT-TO-DELETE and GRADASCENT are poor compared to GNNDELETE because they are not tailored to incorporate the graph structure for unlearning. Regarding the space efficiency, we measure the number of training parameters and show that GNNDELETE has the smallest model size. In addition, the number of training parameters does not scale with respect to the graph size, proving the efficiency and scalability of GNNDELETE. For instance, GNNDELETE takes **9.3×** less computation than GRAPHERASER. We further demonstrate that GNNDELETE can be more efficient by only inserting a deletion operator after the last layer without losing much performance. Additional results and details are in Tables 5-6.

### 5.4 **Q3**: RESULTS – DELETED EDGE CONSISTENCY VS. NEIGHBORHOOD INFLUENCE

We conducted ablations on two key properties of GNNDELETE, namely Deleted Edge Consistency and Neighborhood Influence, by varying the regularization parameter $\lambda$ in Equation 7. The results presented in Table 3 demonstrate that both properties are necessary for achieving high AUROC on both $\mathcal{E}_t$ and $\mathcal{E}_d$. We observed that as $\lambda$ decreases, GNNDELETE focuses more on Neighborhood Influence, which explains why the model's performance on $\mathcal{E}_t$ is close to the original, while it cannot distinguish $\mathcal{E}_d$ from the remaining edges. Conversely, for higher values of $\lambda$, GNNDELETE focuses more on optimizing the Deleted Edge Consistency property and can better distinguish between $\mathcal{E}_d$ and $\mathcal{E}_r$. In summary, we observed a 5.56% improvement in the average AUROC for $\lambda = 0.5$.

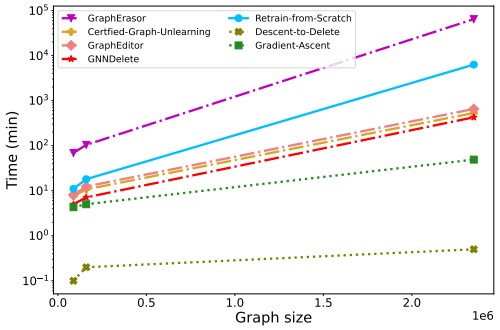

**Table 3:** Ablation study on the interplay of Deleted Edge Consistency and Neighborhood Influence property. **Unlearning task: 2.5% edge deletion. Evaluation: link prediction. Dataset: DBLP**. The gap is calculated as: $|\text{AUROC}(\mathcal{E}_t) - \text{AUROC}(\mathcal{E}_d)|$. Best overall deletion performance is achieved for $\lambda = 0.5$, indicating that both properties are necessary to successfully delete information from the GNN model while minimizing negative effects on overall model performance.

**Figure 2:** Comparison of efficiency on three datasets (PubMed, CS, and OGB-Collab). We plot the retraining approach in solid lines, general unlearning methods in dotted lines, and graph unlearning methods in dash-dotted lines. Results show that GNNDELETE scales better than existing graph unlearning methods, as its execution time is consistently lower than other methods, especially for larger graphs.

| $\lambda$ | AUROC on $\mathcal{E}_t$ | AUROC on $\mathcal{E}_d$ | Avg. AUROC (Gap) |
|---|---|---|---|
| 0.0 | 0.964 $\pm$0.003 | 0.492 $\pm$0.012 | 0.728 (0.473) |
| 0.2 | 0.961 $\pm$0.003 | 0.593 $\pm$0.011 | 0.777 (0.368) |
| 0.4 | 0.950 $\pm$0.005 | 0.691 $\pm$0.010 | 0.821 (0.259) |
| 0.5 | 0.934 $\pm$0.002 | 0.748 $\pm$0.006 | **0.841 (0.185)** |
| 0.6 | 0.927 $\pm$0.001 | 0.739 $\pm$0.006 | 0.834 (0.188) |
| 0.8 | 0.893 $\pm$0.003 | 0.759 $\pm$0.008 | 0.823 (0.134) |
| 1.0 | 0.858 $\pm$0.004 | 0.757 $\pm$0.004 | 0.808 (0.101) |

## 6 CONCLUSION

We introduce GNNDELETE, a novel deletion operator that is both flexible and easy-to-use, and can be applied to any type of graph neural network (GNN) model. We also introduce two properties, denoted as Deleted Edge Consistency and Neighborhood Influence, which can contribute to more effective graph unlearning. By combining the deletion operator with these two properties, we define a novel loss function for graph unlearning. We evaluate GNNDELETE across a wide range of deletion tasks including edge deletion, node deletion, and node feature unlearning, and demonstrate that it outperforms existing graph unlearning models. Our experiments show that GNNDELETE performs consistently well across a variety of tasks and is easy to use. Results demonstrate the potential of GNNDELETE as a general strategy for graph unlearning.

ACKNOWLEDGEMENTS

We gratefully acknowledge the support of the Under Secretary of Defense for Research and Engineering under Air Force Contract No. FA8702-15-D-0001 and awards from Harvard Data Science Initiative, Amazon Research Award, Bayer Early Excellence in Science Award, AstraZeneca Research, and Roche Alliance with Distinguished Scientists Award. G.D. is supported by the Harvard Data Science Initiative Postdoctoral Fellowship. Any opinions, findings, conclusions or recommendations expressed in this material are those of the authors and do not necessarily reflect the views of the funders. The authors declare that there are no conflict of interests.

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

## A    FURTHER DETAILS ON RELATED WORK

**Connection with Catastrophic Forgetting and Dynamic Network Embeddings.** Despite the fact of forgetting knowledge, we argue that catastrophic forgetting (Kemker et al., 2018) is not a good fit for the machine unlearning setting. Specifically, catastrophic forgetting is 1) not targeted at a particular set of data, 2) not guaranteed to forget, i.e. adversarial agents may still find out the presence of what is forgotten in the training set; 3) not capable of handling an arbitrary amount of deletion requests. Dynamic network embedding (Nguyen et al., 2018) learns network representations by incorporating time information and can deal with data deletion. However, these methods are designed to process graphs that are inherently changing over time. On the other hand, machine unlearning aims at deleting a specific set of training data, while keeping the majority of the underlying graph fixed. It aims at targeted data removal requests and is thus orthogonal to the above two approaches.

**Connection with Techniques for Achieving Privacy in ML Models.** Privacy-preserving is another related privacy-preserving learning scheme, such as federated learning (FL) (Konečný et al., 2016). To let FL models unlearn data in a similar privacy-preserving way, Liu et al. (2021) proposed the first federated unlearning framework by leveraging historical parameter updates. Chen et al. (2021c) proposed a new attack and metrics to improve privacy protection, which provides insights on practical implementations of machine unlearning. Golatkar et al. (2021) proposed machine unlearning in a mixed-privacy setting by splitting the weights into a set of core and forgettable user weights. Different from privacy-preserving algorithms that aim at protecting data privacy during training and inference, the goal of machine unlearning, most of the time, is to retrieve privacy, which has no conflict.

**Machine Unlearning for Other Tasks.** Given the increasing necessity of machine unlearning, researches have studied machine unlearning algorithms tailored for other tasks. For example, Chen et al. (2022a) and Li et al. (2022b) propose machine unlearning frameworks for recommendation systems by considering the collaborative information. In addition, recent unlearning methods use Bayesian and latent models (Nguyen et al., 2020; Kassab & Simeone, 2022; Sekhari et al., 2021). Machine unlearning (MU) has also been applied for other tasks, including verification of MU (Sommer et al., 2022), re-definition of MU (Thudi et al., 2022b), test of MU (Goel et al., 2022), zero-shot MU (Chundawat et al., 2022), tree-based MU (Schelter et al., 2021), coded MU (Aldaghri et al., 2021), pruning based federated MU (Wang et al., 2022), adaptive sequence deletion MU (Gupta et al., 2021), MU for k-means (Ginart et al., 2019), MU for features and labels (Warnecke et al., 2023), MU for spam email detection (Parne et al., 2021), MU for linear models (Mahadevan & Mathioudakis, 2021). While the above mentioned methods have shown new directions for machine unlearning, they are not comparable to our work as we focus on general graph unlearning instead of a specific ML task.

## B    PROOF OF THEOREM 1

For the $L$-th GNN layer, we assume for sake of simplicity that the final node representation $z_u$ for a node $u$ is computed as:

$$z_u = \sigma\Big(W_1 h_u^{L-1} + \sum_{v \in \mathcal{N}_u} W_2 \mathbf{h}_v^{L-1}\Big), \tag{8}$$

where $\sigma$ is the sigmoid function, and, and $W_1, W_2$ are the weight parameters for the GNN layer. Similarly, for the unlearned $L$-th layer we have:

$$z'_u = \sigma(W_D^L z_u^L), \tag{9}$$

where $W_D^L$'s are the weight parameters for the DEL operator. For a given edge $e_{uv} \in \mathcal{E}_d$ that is to be deleted, we expect for the dot product $\langle z_u, z_v \rangle$ to be maximized, as it is an existent edge in the initial graph, while $\langle z'_u, z'_v \rangle$ to be fixed, so that the edge probability is close to $0.5$, following the *Deleted Edge Consistency* property. Specifically, the difference of the two terms can be bounded as:

$$\langle z_u, z_v \rangle - \langle z'_u, z'_v \rangle = \frac{1}{2}(\|z_u\|^2 + \|z_v\|^2 - \|z_u - z_v\|^2) - \frac{1}{2}(\|z'_u\|^2 + \|z'_v\|^2 + \|z'_u - z'_v\|^2)$$

$$\langle z_u, z_v \rangle - \langle z'_u, z'_v \rangle \overset{\text{Normalization}}{=} 1 - \frac{1}{2}\|z_u - z_v\|^2 - 1 - \frac{1}{2}\|z'_u - z'_v\|^2$$

$$\tag{10}$$

Then, simplifying Equations 8 and 9, we have:

$$\|\boldsymbol{z}_u' - \boldsymbol{z}_v'\| = \|\sigma(\boldsymbol{W}_D^L \boldsymbol{z}_u) - \sigma(\boldsymbol{W}_D^L \boldsymbol{z}_v)\|$$

$$\overset{\text{Lipschitz } \sigma}{\leq} \|\boldsymbol{W}_D^L \boldsymbol{z}_u - \boldsymbol{W}_D^L \boldsymbol{z}_v\|$$

$$\overset{\text{Cauchy-Schwartz}}{\leq} \|\boldsymbol{W}_D^L\|\|\boldsymbol{z}_u - \boldsymbol{z}_v\|$$

and applying that to Equation 10:

$$\langle \boldsymbol{z}_u, \boldsymbol{z}_v \rangle - \langle \boldsymbol{z}_u', \boldsymbol{z}_v' \rangle \geq -\frac{1}{2}\|\boldsymbol{z}_u - \boldsymbol{z}_v\|^2 - \frac{1}{2}\|\boldsymbol{W}_D^L\|^2\|\boldsymbol{z}_u - \boldsymbol{z}_v\|^2$$

$$\langle \boldsymbol{z}_u, \boldsymbol{z}_v \rangle - \langle \boldsymbol{z}_u', \boldsymbol{z}_v' \rangle \geq -\frac{1}{2}(1 + \|\boldsymbol{W}_D^L\|^2)\|\boldsymbol{z}_u - \boldsymbol{z}_v\|^2$$

$$\tag{11}$$

## C  EXPERIMENTS

In this section, we give further details on the experimental setup. We firstly present the used evaluation metrics for the edge and node deletion, then we describe the sampling strategies for $\mathcal{E}_t$, and $\mathcal{E}_d$, dataset statistics, and finally we present the full results for the model efficiency and the prediction performance of the models.

### C.1  EVALUATION METRICS

We choose the following performance metrics to measure the effectiveness of the deletion operation on the set of the deleted edges $\mathcal{E}_d$ and the test set $\mathcal{E}_t$ that contains a subset of the remaining edges:. For the edge deletion case, we have:

- AUROC and AUPRC on the test set $\mathcal{E}_t$: these metrics measure the prediction perfomance of each graph graph learning model over the existent edges of the test set $\mathcal{E}_t$. High values of AUROC and AUPRC on $\mathcal{E}_t$ show that the unlearned model's performance on the original test set is not affected by the deletion of an edge subset.
- AUROC and AUPRC on the set of deleted edges $\mathcal{E}_d$: these metrics quantify the ability of the unlearned models to distinguish deleted edges (contained in $\mathcal{E}_d$) from the remaining edges (contained in $\mathcal{E}_r$). For the computation of the area under the curve, we take into account the total of the deleted edges in $\mathcal{E}_d$ and we sample an equal amount of remaining edges from $\mathcal{E}_r$. Then, we set the labels of the deleted edges equal to $0$ (since they do not exist after deletion), and the labels of the remaining edges to $1$. Higher values of the AUC on $\mathcal{E}_d$ show that the model is more capable of distinguishing edges in $\mathcal{E}_d$ from edges in $\mathcal{E}_r$.

For the node deletion case, we capitalize on the evaluation for privacy leakage with Membership Inference (MI) (Thudi et al., 2022a) attacks. An unlearned model $f'$ effectively forgets $\mathcal{E}_d$ if an MI attacker returns that $\mathcal{E}_d$ is not present in the training set, i.e., the probability of predicting the presence of $\mathcal{E}_d$ decreases.

- MI Ratio: this metric quantifies the success rate of a Membership Inference (MI) attack, by calculating the ratio of presence probability of $\mathcal{E}_d$ before and after the deletion operator. We adapt the implementation from Olatunji et al. (2021) in our experiments. If the ratio is higher than $1$, it means that the model contains less information about $\mathcal{E}_d$. If the ratio is less than $1$, it means the model contains more information about $\mathcal{E}_d$.

### C.2  SAMPLING OF EDGES IN $\mathcal{E}_t$ AND $\mathcal{E}_d$

We sample $5\%$ of the total edges as the test set ($\mathcal{E}_t$) to evaluate the model's performance on link prediction and sample another $5\%$ as validation set for selecting the best model. We propose two sampling strategies for sampling $\mathcal{E}_d$ for edge deletion tasks: i) $\mathcal{E}_{d,\text{OUT}}$ refers to randomly sampling edges outside the 2-hop enclosing subgraph of $\mathcal{E}_t$, i.e., $\mathcal{E}_{d,\text{OUT}} = \{e | e \notin S_{\mathcal{E}_t}^2\}$; and ii) $\mathcal{E}_{d,\text{IN}}$ refers to randomly sampling edges from the 2-hop enclosing subgraph of $\mathcal{E}_t$, i.e., $\mathcal{E}_{d,\text{IN}} = \{e | e \in S_{\mathcal{E}_t}^2\}$. We note that deleting $\mathcal{E}_{d,\text{IN}}$ is more difficult than deleting $\mathcal{E}_{d,\text{OUT}}$ as the deletion operation will have

an impact on the local neighborhood of $\mathcal{E}_{d,\text{IN}}$, where $\mathcal{E}_t$ is located. For comparison of these two sampling strategies, please refer to the Appendix (Tables 13-14) fore results on Cora, Tables 15-16 for results on PubMed, Tables 17-18 on DBLP, Tables 19-20 on OGB-Collab, and Tables 21-22 on WordNet18.

## C.3 EVALUATED DATASETS

Table 4 presents the size of the graphs and the maximum number of deleted edges in the experiments, ranging from small to large scales.

**Table 4:** Statistics of evaluated datasets and a maximum number of deleted edges.

| Graph | # Nodes | # Edges | # Unique edge types | Max # deleted edges |
|---|---|---|---|---|
| Cora | 19,793 | 126,842 | 1 | 6,342 |
| PubMed | 19,717 | 88,648 | 1 | 4,432 |
| DBLP | 17,716 | 105,734 | 1 | 5,286 |
| CS | 18,333 | 163,788 | 1 | 8,189 |
| OGB-Collab | 235,868 | 2,358,104 | 1 | 117,905 |
| WordNet18 | 40,943 | 151,442 | 18 | 7,072 |
| OGB-BioKG | 93,773 | 5,088,434 | 51 | 127,210 |

## C.4 MODEL EFFICIENCY

**Space efficiency.** Space efficiency is reflected by the number of trainable parameters a model has. We report number of parameters for all methods in Table 5. We can observe that GNNDELETE has the smallest model size, which does not scale with respect to the size of the graph. This proves the efficiency and scalability of GNNDELETE. It is also significantly smaller than GRAPHERASER, where we usually have to divide the original graph into 10 or 20 shards, each requiring a separate GNN model.

**Table 5:** Space efficiency of unlearning models. All models are trained on OGB-Collab and OGB-BioKG to delete 5.0% of edges, using a 2-layer GCN/R-GCN architecture with 128, 64, 32 as hidden dimensions, with the number of shards in GRAPHERASER as 10. Shown is the number of trainable parameters in each deletion model.

| Model | OGB-Collab | OGB-BioKG |
|---|---|---|
| RETRAIN | 5,216 | 12,009,792 |
| GRADASCENT | 5,216 | 12,009,792 |
| D2D | 5,216 | 12,009,792 |
| GRAPHERASER | 52,160 | 120,097,920 |
| GRAPHEDITOR | 5,216 | N/A |
| CERTUNLEARN | 5,216 | N/A |
| GNNDELETE | 5,120 | 5,120 |

**Layer-wise unlearning vs. Last layer-only unlearning.** Even though GNNDELETE achieves good efficiency, it may still be expensive when deleting a large number of nodes/edges. GNNDELETE can function under such extreme conditions and alleviate the computation cost by adopting a *Last layer-only* strategy, where a single deletion operator is inserted after the last layer. As shown in Table 6, *last layer-only* GNNDELETE does not lead to significant performance degradation, with only 0.7% and 0.9% drop on link prediction performance on $\mathcal{E}_t$ and $\mathcal{E}_d$, respectively. While existing unlearning models don't have such flexibility to run in an lightweight fashion.

## C.5 EVALUATIONS ON OTHER DOWNSTREAM TASKS

We recognize that the deleted information has an influence on many downstream tasks. For example, removing edges impacts node classification performance as well. We evaluate unlearning methods on three canonical downstream tasks: 1) node classification, 2) link prediction, and 3) graph classification.

**Table 6:** Comparison between *layer-wise* and *last layer-only* GNNDELETE. **Unlearning task: 2.5% edge deletion on DBLP. Evaluated on: link prediction**. Performance is in AUROC ($\uparrow$).

| Model | $\mathcal{E}_t$ | $\mathcal{E}_d$ |
|---|---|---|
| RETRAIN | 0.964 $\pm$0.003 | 0.506 $\pm$0.013 |
| GNNDELETE - layer-wise | 0.934 $\pm$0.002 | 0.748 $\pm$0.006 |
| GNNDELETE - last layer-only | 0.927 $\pm$0.005 | 0.739 $\pm$0.005 |

In addition to link prediction evaluation in Table 1, we summarize the performance on other two downstream tasks in Table 7.

**Table 7: Unlearning task: 2.5% edge deletion. Evaluated on: node classification on DBLP (Accuracy ($\uparrow$)) and graph classification on ogbg-molhiv (AUROC ($\uparrow$)).** The best performance is marked in **bold** and the second best in nderline.

| Model | Node classification | Graph classification |
|---|---|---|
| RETRAIN | 0.810 $\pm$0.021 | 0.758 $\pm$0.068 |
| GRADASCENT | 0.614 $\pm$0.042 | 0.601 $\pm$0.063 |
| D2D | 0.250 $\pm$0.000 | 0.572 $\pm$0.013 |
| GRAPHERASER | 0.682 $\pm$0.044 | 0.596 $\pm$0.032 |
| GRAPHEDITOR | 0.711 $\pm$0.039 | 0.613 $\pm$0.035 |
| CERTUNLEARN | 0.739 $\pm$0.024 | 0.607 $\pm$0.028 |
| GNNDELETE | **0.782 $\pm$0.027** | **0.710 $\pm$0.041** |

## C.6 NODE DELETION

GNNDELETE can be applied to unlearn nodes in a graph by optimizing the same loss function in Eq. 7. Table 8 presents the results of randomly deleting 100 nodes on DBLP dataset. In addition to the standard node classification evaluation, we also present the performance on link prediction and graph classification.

**Table 8: Unlearning task: 100 node deletion. Evaluated on: node classification, link prediction (AUROC ($\uparrow$)). Dataset: DBLP.** The best performance is marked in **bold** and the second best in underline.

| Model | Accuracy | F1 | MI ratio | Link prediction |
|---|---|---|---|---|
| RETRAIN | 0.845 $\pm$0.008 | 0.841 $\pm$0.004 | 1.515 $\pm$0.034 | 0.973 $\pm$0.002 |
| GRADASCENT | 0.392 $\pm$0.026 | 0.341 $\pm$0.035 | 1.021 $\pm$0.113 | 0.571 $\pm$0.032 |
| D2D | 0.250 $\pm$0.000 | 0.250 $\pm$0.000 | **1.755 $\pm$0.065** | 0.507 $\pm$0.002 |
| GRAPHERASER | 0.718 $\pm$0.014 | 0.716 $\pm$0.011 | 0.975 $\pm$0.083 | 0.513 $\pm$0.004 |
| GRAPHEDITOR | 0.765 $\pm$0.012 | 0.749 $\pm$0.006 | 1.260 $\pm$0.088 | 0.697 $\pm$0.031 |
| CERTUNLEARN | 0.743 $\pm$0.027 | 0.738 $\pm$0.022 | 1.134 $\pm$0.009 | 0.713 $\pm$0.025 |
| GNNDELETE | **0.793 $\pm$0.016** | **0.768 $\pm$0.009** | 1.401 $\pm$0.082 | **0.938 $\pm$0.004** |

## C.7 NODE FEATURE DELETION

GNNDELETE can be applied to update node features by optimizing the same loss function in Eq. 7. For node and edge level evaluation, we randomly select 100 nodes and update $X_d = 0$. For graph classification evaluation, we randomly choose 100 graphs and update $X_d = 0$.

## C.8 SEQUENTIAL UNLEARNING

**Problem Formulation 2 (Sequential Graph Unlearning).** Given a graph $G = (\mathcal{V}, \mathcal{E}, \mathbf{X})$, a fully trained GNN model $m(G)$, and a sequence of edges $\mathcal{E}_{seq,d} = \{\mathcal{E}_{d,0}, \mathcal{E}_{d,1}, \ldots, \mathcal{E}_{d,S}\}, |\mathcal{E}_{seq,d}| = S$, where each deletion request $\mathcal{E}_{d,i} \in \mathcal{E}_{seq,d}$ is a standard graph unlearning problem (as defined in Sec. 4). Then, sequential graph unlearning aims to unlearn every batch of edges $\mathcal{E}_{d,i} \in \mathcal{E}_{seq,d}$ from the GNN model $m(G)$ in a **sequential** manner meaning that a request $\mathcal{E}_{d,i+1}$ is given to the unlearned model after $\mathcal{E}_{d,i}$ has been successfully unlearned.

**Table 9: Unlearning task: 100 node feature update. Evaluated on: node classification on DBLP (Acc. (↑)), link prediction on DBLP (AUROC (↑)), and graph classification on ogbg-molhiv (AUROC (↑)). Bold**: best performance. Underline: second best performance.

| Model | Node classification | Link prediction | Graph classification |
|---|---|---|---|
| RETRAIN (reference only) | 0.810 ±0.021 | 0.897 ±0.025 | 0.753 ±0.042 |
| GRADASCENT | 0.614 ±0.042 | 0.655 ±0.030 | 0.623 ±0.046 |
| D2D | 0.525 ±0.009 | 0.504 ±0.003 | 0.586 ±0.017 |
| GRAPHERASER | 0.682 ±0.044 | 0.579 ±0.021 | 0.592 ±0.032 |
| GRAPHEDITOR | 0.711 ±0.039 | 0.683 ±0.031 | 0.630 ±0.026 |
| CERTUNLEARN | 0.739 ±0.024 | 0.667 ±0.019 | 0.687 ±0.030 |
| GNNDELETE | **0.782 ±0.027** | **0.850 ±0.027** | **0.724 ±0.027** |

GNNDelete is designed to handle a sequence of deletion requests. In stark contrast to existing deletion techniques (GraphEraser (Chen et al., 2022b), GraphEditor (Cong & Mahdavi, 2023), Descent-to-Delete (Neel et al., 2021)), GNNDelete does not require retraining from scratch for additional incoming deletion requests. It achieves sequential unlearning by continuing training the same deletion operator. As $\mathcal{E}_{d,i} \neq \mathcal{E}_{d,i+1}$, we can maintain a binary mask to easily turn on and off the deletion operator for specific nodes following the definition in Eq. 3. This is specified by the binary mask containing information about what node representations are to be updated in the DEL operator.

**Table 10: Unlearning task: 2.5% sequential edge deletion, with a batch size of 0.5%. Evaluated on: link prediction. Dataset: DBLP**. Performance is shown in AUROC (↑) on GCN architecture.

| Ratio (%) | $\mathcal{E}_t$ | $\mathcal{E}_d$ |
|---|---|---|
| 0.5 | 0.951 | 0.829 |
| 1.0 | 0.949 | 0.808 |
| 1.5 | 0.943 | 0.791 |
| 2.0 | 0.938 | 0.776 |
| 2.5 | 0.934 | 0.748 |

## C.9 COMPARISON WITH DYNAMIC NETWORK EMBEDDING (DNE)

Graph unlearning can be formulated as a special case of dynamic network. Despite that, we argue that they are fundamentally different. Graph unlearning usually refers to deleting a small part of the graph, while the majority of the graph remains stable. We want a graph unlearning algorithm to selectively forget something it has captured during training. On the contrary, dynamic networks are intrinsically changing over time. The goal of DNE methods is to learn such evolution.

As shown in Table 11, DNE methods are not applicable to graph unlearning task out of the box, emphasizing the need for algorithms designed for graph unlearning.

**Table 11:** Comparison of Graph Unlearning and Dynamic Network Embedding (DNE). **Unlearning task: 2.5% edge deletion. Evaluated on: link prediction. Dataset: DBLP**. Performance is in AUROC (↑).

| Model | $\mathcal{E}_t$ | $\mathcal{E}_d$ |
|---|---|---|
| RETRAIN (reference only) | 0.964 ±0.003 | 0.506 ±0.013 |
| JODIE (DNE) | 0.801 ±0.042 | 0.613 ±0.026 |
| GNNDELETE (Unlearning) | 0.934 ±0.002 | 0.748 ±0.006 |

## C.10 EVALUATE NODE EMBEDDINGS ON OUTLIER DETECTION

## D ADDITIONAL DETAILS ON RESULTS

We detail all the results from different graphs, GNN architectures, models, and deletion ratios.

**Table 12: Unlearning task: 2.5% edge deletion on DBLP. Evaluated on: outlier detection**. We compute the percentage (↑) of edges that are classified as regular edges, i.e., non-outliers on GCN architecture.

| Model | Percentage |
|---|---|
| RETRAIN (reference only) | 0.746 |
| GRADASCENT | 0.503 |
| D2D | 0.517 |
| GRAPHERASER | 0.563 |
| GRAPHEDITOR | 0.642 |
| CERTUNLEARN | 0.625 |
| GNNDELETE | **0.710** |

**Table 13: Unlearning task: edge deletion when $\mathcal{E}_d = \mathcal{E}_{d,\text{OUT}}$. Evaluation: link prediction. Dataset: Cora.** Reported is AUROC (↑) performance with the best marked in **bold** and the second best underlined.

| Ratio (%) | Model | GCN | | GAT | | GIN | |
|---|---|---|---|---|---|---|---|
| | | $\mathcal{E}_t$ | $\mathcal{E}_d$ | $\mathcal{E}_t$ | $\mathcal{E}_d$ | $\mathcal{E}_t$ | $\mathcal{E}_d$ |
| 0.5 | RETRAIN | 0.965 ±0.002 | 0.783 ±0.018 | 0.961 ±0.002 | 0.756 ±0.013 | 0.961 ±0.002 | 0.815 ±0.015 |
| | GRADASCENT | 0.536 ±0.010 | 0.618 ±0.014 | 0.517 ±0.017 | 0.558 ±0.034 | 0.751 ±0.049 | 0.778 ±0.043 |
| | D2D | 0.500 ±0.000 | 0.500 ±0.000 | 0.500 ±0.000 | 0.500 ±0.000 | 0.500 ±0.000 | 0.500 ±0.000 |
| | GRAPHERASER | 0.563 ±0.002 | 0.500 ±0.000 | 0.553 ±0.013 | 0.500 ±0.000 | 0.554 ±0.009 | 0.500 ±0.000 |
| | GRAPHEDITOR | 0.805 ±0.077 | 0.614 ±0.054 | - | - | - | - |
| | CERTUNLEARN | 0.814 ±0.065 | 0.603 ±0.039 | - | - | - | - |
| | GNNDELETE | **0.958 ±0.002** | **0.977 ±0.001** | **0.953 ±0.002** | **0.979 ±0.001** | **0.956 ±0.003** | **0.953 ±0.010** |
| 2.5 | RETRAIN | 0.966 ±0.001 | 0.790 ±0.009 | 0.961 ±0.002 | 0.758 ±0.011 | 0.961 ±0.003 | 0.833 ±0.010 |
| | GRADASCENT | 0.504 ±0.002 | 0.494 ±0.004 | 0.510 ±0.019 | 0.522 ±0.023 | 0.603 ±0.039 | 0.605 ±0.030 |
| | D2D | 0.500 ±0.000 | 0.500 ±0.000 | 0.500 ±0.000 | 0.500 ±0.000 | 0.500 ±0.000 | 0.500 ±0.000 |
| | GRAPHERASER | 0.542 ±0.002 | 0.500 ±0.000 | 0.519 ±0.013 | 0.500 ±0.000 | 0.563 ±0.009 | 0.500 ±0.000 |
| | GRAPHEDITOR | 0.754 ±0.023 | 0.583 ±0.056 | - | - | - | - |
| | CERTUNLEARN | 0.795 ±0.037 | 0.578 ±0.015 | - | - | - | - |
| | GNNDELETE | **0.953 ±0.002** | **0.912 ±0.004** | **0.949 ±0.003** | **0.914 ±0.004** | **0.953 ±0.002** | **0.922 ±0.006** |
| 5.0 | RETRAIN | 0.966 ±0.002 | 0.812 ±0.006 | 0.961 ±0.001 | 0.778 ±0.006 | 0.960 ±0.003 | 0.852 ±0.006 |
| | GRADASCENT | 0.557 ±0.122 | 0.513 ±0.106 | 0.520 ±0.042 | 0.517 ±0.036 | 0.580 ±0.027 | 0.572 ±0.018 |
| | D2D | 0.500 ±0.000 | 0.500 ±0.000 | 0.500 ±0.000 | 0.500 ±0.000 | 0.500 ±0.000 | 0.500 ±0.000 |
| | GRAPHERASER | 0.514 ±0.002 | 0.500 ±0.000 | 0.523 ±0.013 | 0.500 ±0.000 | 0.533 ±0.009 | 0.500 ±0.000 |
| | GRAPHEDITOR | 0.721 ±0.048 | 0.545 ±0.056 | - | - | - | - |
| | CERTUNLEARN | 0.745 ±0.033 | 0.513 ±0.012 | - | - | - | - |
| | GNNDELETE | **0.953 ±0.003** | **0.882 ±0.005** | **0.951 ±0.002** | **0.872 ±0.004** | **0.950 ±0.003** | **0.914 ±0.004** |

**Table 14: Unlearning task: edge deletion when $\mathcal{E}_d = \mathcal{E}_{d,\text{IN}}$. Evaluation: link prediction. Dataset: Cora.** Reported is AUROC (↑) performance with the best marked in **bold** and the second best underlined.

| Ratio (%) | Model | GCN | | GAT | | GIN | |
|---|---|---|---|---|---|---|---|
| | | $\mathcal{E}_t$ | $\mathcal{E}_d$ | $\mathcal{E}_t$ | $\mathcal{E}_d$ | $\mathcal{E}_t$ | $\mathcal{E}_d$ |
| 0.5 | RETRAIN | 0.965 ±0.002 | 0.511 ±0.024 | 0.961 ±0.001 | 0.513 ±0.024 | 0.960 ±0.003 | 0.571 ±0.028 |
| | GRADASCENT | 0.528 ±0.008 | 0.588 ±0.014 | 0.502 ±0.002 | 0.543 ±0.058 | 0.792 ±0.046 | 0.705 ±0.113 |
| | D2D | 0.500 ±0.000 | 0.500 ±0.000 | 0.500 ±0.000 | 0.500 ±0.000 | 0.500 ±0.000 | 0.500 ±0.000 |
| | GRAPHERASER | 0.528 ±0.002 | 0.500 ±0.000 | 0.523 ±0.013 | 0.500 ±0.000 | 0.542 ±0.009 | 0.500 ±0.000 |
| | GRAPHEDITOR | 0.704 ±0.057 | 0.488 ±0.024 | - | - | - | - |
| | CERTUNLEARN | 0.811 ±0.035 | 0.497 ±0.013 | - | - | - | - |
| | GNNDELETE | **0.944 ±0.003** | **0.843 ±0.015** | **0.937 ±0.004** | **0.880 ±0.011** | **0.942 ±0.005** | **0.824 ±0.021** |
| 2.5 | RETRAIN | 0.966 ±0.002 | 0.520 ±0.008 | 0.961 ±0.001 | 0.520 ±0.012 | 0.958 ±0.002 | 0.583 ±0.007 |
| | GRADASCENT | 0.509 ±0.006 | 0.509 ±0.007 | 0.490 ±0.007 | 0.551 ±0.014 | 0.639 ±0.077 | 0.614 ±0.016 |
| | D2D | 0.500 ±0.000 | 0.500 ±0.000 | 0.500 ±0.000 | 0.500 ±0.000 | 0.500 ±0.000 | 0.500 ±0.000 |
| | GRAPHERASER | 0.517 ±0.002 | 0.500 ±0.000 | 0.556 ±0.013 | 0.500 ±0.000 | 0.547 ±0.009 | 0.500 ±0.000 |
| | GRAPHEDITOR | 0.673 ±0.091 | 0.493 ±0.027 | - | - | - | - |
| | CERTUNLEARN | 0.781 ±0.042 | 0.492 ±0.015 | - | - | - | - |
| | GNNDELETE | **0.925 ±0.006** | **0.716 ±0.003** | **0.928 ±0.007** | **0.738 ±0.005** | **0.919 ±0.004** | **0.745 ±0.005** |
| 5.0 | RETRAIN | 0.964 ±0.002 | 0.525 ±0.008 | 0.960 ±0.001 | 0.525 ±0.007 | 0.958 ±0.002 | 0.591 ±0.006 |
| | GRADASCENT | 0.509 ±0.005 | 0.487 ±0.003 | 0.489 ±0.015 | 0.537 ±0.007 | 0.592 ±0.031 | 0.583 ±0.013 |
| | D2D | 0.500 ±0.000 | 0.500 ±0.000 | 0.500 ±0.000 | 0.500 ±0.000 | 0.500 ±0.000 | 0.500 ±0.000 |
| | GRAPHERASER | 0.528 ±0.002 | 0.500 ±0.000 | 0.517 ±0.013 | 0.500 ±0.000 | 0.530 ±0.009 | 0.500 ±0.000 |
| | GRAPHEDITOR | 0.587 ±0.014 | 0.475 ±0.015 | - | - | - | - |
| | CERTUNLEARN | 0.664 ±0.023 | 0.457 ±0.021 | - | - | - | - |
| | GNNDELETE | **0.916 ±0.007** | **0.680 ±0.006** | **0.920 ±0.005** | **0.700 ±0.004** | **0.900 ±0.005** | **0.717 ±0.003** |

**Table 15: Unlearning task: edge deletion when $\mathcal{E}_d = \mathcal{E}_{d,\textbf{OUT}}$. Evaluation: link prediction. Dataset: PubMed.** Reported is AUROC ($\uparrow$) performance with the best marked in **bold** and the second best underlined.

| Ratio (%) | Model | GCN $\mathcal{E}_t$ | GCN $\mathcal{E}_d$ | GAT $\mathcal{E}_t$ | GAT $\mathcal{E}_d$ | GIN $\mathcal{E}_t$ | GIN $\mathcal{E}_d$ |
|---|---|---|---|---|---|---|---|
| 0.5 | RETRAIN | 0.968 ±0.001 | 0.687 ±0.023 | 0.931 ±0.003 | 0.723 ±0.026 | 0.941 ±0.004 | 0.865 ±0.012 |
| | GRADASCENT | 0.458 ±0.139 | 0.539 ±0.091 | 0.450 ±0.017 | 0.541 ±0.049 | 0.518 ±0.122 | 0.528 ±0.021 |
| | D2D | 0.500 ±0.000 | 0.500 ±0.000 | 0.500 ±0.000 | 0.500 ±0.000 | 0.500 ±0.000 | 0.500 ±0.000 |
| | GRAPHERASER | 0.529 ±0.013 | 0.500 ±0.000 | 0.542 ±0.004 | 0.500 ±0.000 | 0.535 ±0.003 | 0.500 ±0.000 |
| | GRAPHEDITOR | 0.732 ±0.043 | 0.603 ±0.015 | - | - | - | - |
| | CERTUNLEARN | 0.724 ±0.012 | 0.597 ±0.029 | - | - | - | - |
| | GNNDELETE | **0.961 ±0.004** | **0.973 ±0.005** | **0.926 ±0.006** | **0.976 ±0.005** | **0.940 ±0.005** | **0.963 ±0.010** |
| 2.5 | RETRAIN | 0.967 ±0.001 | 0.696 ±0.011 | 0.930 ±0.003 | 0.736 ±0.011 | 0.942 ±0.005 | 0.875 ±0.008 |
| | GRADASCENT | 0.446 ±0.130 | 0.500 ±0.067 | 0.582 ±0.006 | 0.758 ±0.033 | 0.406 ±0.054 | 0.454 ±0.037 |
| | D2D | 0.500 ±0.000 | 0.500 ±0.000 | 0.500 ±0.000 | 0.500 ±0.000 | 0.500 ±0.000 | 0.500 ±0.000 |
| | GRAPHERASER | 0.505 ±0.024 | 0.500 ±0.000 | 0.538 ±0.009 | 0.500 ±0.000 | 0.544 ±0.014 | 0.500 ±0.000 |
| | GRAPHEDITOR | 0.689 ±0.015 | 0.570 ±0.011 | - | - | - | - |
| | CERTUNLEARN | 0.697 ±0.012 | 0.582 ±0.032 | - | - | - | - |
| | GNNDELETE | **0.954 ±0.003** | **0.909 ±0.004** | **0.920 ±0.004** | **0.916 ±0.006** | **0.943 ±0.005** | **0.938 ±0.009** |
| 5.0 | RETRAIN | 0.966 ±0.001 | 0.707 ±0.004 | 0.929 ±0.002 | 0.744 ±0.008 | 0.942 ±0.004 | 0.885 ±0.010 |
| | GRADASCENT | 0.446 ±0.126 | 0.492 ±0.064 | 0.581 ±0.010 | 0.704 ±0.022 | 0.388 ±0.056 | 0.455 ±0.028 |
| | D2D | 0.500 ±0.000 | 0.500 ±0.000 | 0.500 ±0.000 | 0.500 ±0.000 | 0.500 ±0.000 | 0.500 ±0.000 |
| | GRAPHERASER | 0.532 ±0.001 | 0.500 ±0.000 | 0.527 ±0.022 | 0.500 ±0.000 | 0.524 ±0.015 | 0.500 ±0.000 |
| | GRAPHEDITOR | 0.598 ±0.023 | 0.530 ±0.006 | - | - | - | - |
| | CERTUNLEARN | 0.643 ±0.031 | 0.534 ±0.020 | - | - | - | - |
| | GNNDELETE | **0.950 ±0.003** | **0.859 ±0.005** | **0.921 ±0.005** | **0.863 ±0.006** | **0.941 ±0.002** | **0.930 ±0.009** |

**Table 16: Unlearning task: edge deletion when $\mathcal{E}_d = \mathcal{E}_{d,\textbf{IN}}$. Evaluation: link prediction. Dataset: PubMed.** Reported is AUROC ($\uparrow$) performance with the best marked in **bold** and the second best underlined.

| Ratio (%) | Model | GCN $\mathcal{E}_t$ | GCN $\mathcal{E}_d$ | GAT $\mathcal{E}_t$ | GAT $\mathcal{E}_d$ | GIN $\mathcal{E}_t$ | GIN $\mathcal{E}_d$ |
|---|---|---|---|---|---|---|---|
| 0.5 | RETRAIN | 0.968 ±0.001 | 0.493 ±0.040 | 0.931 ±0.003 | 0.533 ±0.037 | 0.940 ±0.002 | 0.626 ±0.041 |
| | GRADASCENT | 0.469 ±0.095 | 0.496 ±0.058 | 0.436 ±0.028 | 0.553 ±0.029 | 0.687 ±0.060 | 0.556 ±0.042 |
| | D2D | 0.500 ±0.000 | 0.500 ±0.000 | 0.500 ±0.000 | 0.500 ±0.000 | 0.500 ±0.000 | 0.500 ±0.000 |
| | GRAPHERASER | 0.547 ±0.004 | 0.500 ±0.000 | 0.536 ±0.000 | 0.500 ±0.000 | 0.524 ±0.002 | 0.500 ±0.000 |
| | GRAPHEDITOR | 0.669 ±0.005 | 0.469 ±0.021 | - | - | - | - |
| | CERTUNLEARN | 0.657 ±0.015 | 0.515 ±0.027 | - | - | - | - |
| | GNNDELETE | **0.951 ±0.005** | **0.838 ±0.014** | **0.909 ±0.003** | **0.888 ±0.016** | **0.929 ±0.006** | **0.835 ±0.006** |
| 2.5 | RETRAIN | 0.968 ±0.001 | 0.499 ±0.019 | 0.931 ±0.002 | 0.541 ±0.013 | 0.937 ±0.004 | 0.614 ±0.015 |
| | GRADASCENT | 0.470 ±0.087 | 0.474 ±0.039 | 0.522 ±0.066 | 0.704 ±0.086 | 0.631 ±0.050 | 0.499 ±0.018 |
| | D2D | 0.500 ±0.000 | 0.500 ±0.000 | 0.500 ±0.000 | 0.500 ±0.000 | 0.500 ±0.000 | 0.500 ±0.000 |
| | GRAPHERASER | 0.538 ±0.003 | 0.500 ±0.000 | 0.521 ±0.003 | 0.500 ±0.000 | 0.533 ±0.010 | 0.500 ±0.000 |
| | GRAPHEDITOR | 0.657 ±0.006 | 0.467 ±0.006 | - | - | - | - |
| | CERTUNLEARN | 0.622 ±0.009 | 0.468 ±0.025 | - | - | - | - |
| | GNNDELETE | **0.920 ±0.014** | **0.739 ±0.010** | **0.891 ±0.005** | **0.759 ±0.012** | **0.909 ±0.005** | **0.782 ±0.013** |
| 5.0 | RETRAIN | 0.967 ±0.001 | 0.503 ±0.009 | 0.929 ±0.003 | 0.545 ±0.005 | 0.936 ±0.005 | 0.621 ±0.003 |
| | GRADASCENT | 0.473 ±0.090 | 0.473 ±0.038 | 0.525 ±0.069 | 0.686 ±0.090 | 0.635 ±0.073 | 0.493 ±0.018 |
| | D2D | 0.500 ±0.000 | 0.500 ±0.000 | 0.500 ±0.000 | 0.500 ±0.000 | 0.500 ±0.000 | 0.500 ±0.000 |
| | GRAPHERASER | 0.551 ±0.004 | 0.500 ±0.000 | 0.524 ±0.020 | 0.500 ±0.000 | 0.531 ±0.000 | 0.500 ±0.000 |
| | GRAPHEDITOR | 0.556 ±0.007 | 0.468 ±0.002 | - | - | - | - |
| | CERTUNLEARN | 0.572 ±0.013 | 0.477 ±0.028 | - | - | - | - |
| | GNNDELETE | **0.916 ±0.006** | **0.691 ±0.012** | **0.887 ±0.009** | **0.713 ±0.005** | **0.895 ±0.004** | **0.761 ±0.005** |

**Table 17: Unlearning task: edge deletion when $\mathcal{E}_d = \mathcal{E}_{d,\text{OUT}}$. Evaluation: link prediction. Dataset: DBLP.** Reported is AUROC ($\uparrow$) performance with the best marked in **bold** and the second best underlined.

| Ratio (%) | Model | GCN $\mathcal{E}_t$ | $\mathcal{E}_d$ | GAT $\mathcal{E}_t$ | $\mathcal{E}_d$ | GIN $\mathcal{E}_t$ | $\mathcal{E}_d$ |
|---|---|---|---|---|---|---|---|
| 0.5 | RETRAIN | 0.965 ±0.002 | 0.783 ±0.018 | 0.956 ±0.002 | 0.744 ±0.021 | 0.934 ±0.003 | 0.861 ±0.019 |
| | GRADASCENT | 0.567 ±0.008 | 0.696 ±0.017 | 0.501 ±0.030 | 0.667 ±0.052 | 0.753 ±0.055 | 0.789 ±0.091 |
| | D2D | 0.500 ±0.000 | 0.500 ±0.000 | 0.500 ±0.000 | 0.500 ±0.000 | 0.500 ±0.000 | 0.500 ±0.000 |
| | GRAPHERASER | 0.518 ±0.002 | 0.500 ±0.000 | 0.523 ±0.013 | 0.500 ±0.000 | 0.517 ±0.009 | 0.500 ±0.000 |
| | GRAPHEDITOR | 0.790 ±0.032 | 0.624 ±0.017 | - | - | - | - |
| | CERTUNLEARN | 0.763 ±0.025 | 0.604 ±0.022 | - | - | - | - |
| | GNNDELETE | **0.959 ±0.002** | **0.964 ±0.005** | **0.950 ±0.002** | **0.980 ±0.003** | **0.924 ±0.006** | **0.894 ±0.020** |
| 2.5 | RETRAIN | 0.965 ±0.002 | 0.777 ±0.009 | 0.955 ±0.003 | 0.739 ±0.005 | 0.934 ±0.003 | 0.858 ±0.002 |
| | GRADASCENT | 0.528 ±0.015 | 0.583 ±0.016 | 0.501 ±0.026 | 0.576 ±0.017 | 0.717 ±0.022 | 0.766 ±0.019 |
| | D2D | 0.500 ±0.000 | 0.500 ±0.000 | 0.500 ±0.000 | 0.500 ±0.000 | 0.500 ±0.000 | 0.500 ±0.000 |
| | GRAPHERASER | 0.515 ±0.002 | 0.500 ±0.000 | 0.563 ±0.013 | 0.500 ±0.000 | 0.552 ±0.009 | 0.500 ±0.000 |
| | GRAPHEDITOR | 0.769 ±0.040 | 0.607 ±0.017 | - | - | - | - |
| | CERTUNLEARN | 0.747 ±0.033 | 0.616 ±0.019 | - | - | - | - |
| | GNNDELETE | **0.957 ±0.003** | **0.892 ±0.004** | **0.949 ±0.003** | **0.905 ±0.002** | **0.926 ±0.007** | **0.898 ±0.017** |
| 5.0 | RETRAIN | 0.964 ±0.003 | 0.788 ±0.006 | 0.955 ±0.003 | 0.748 ±0.008 | 0.936 ±0.004 | 0.868 ±0.005 |
| | GRADASCENT | 0.555 ±0.099 | 0.591 ±0.065 | 0.501 ±0.023 | 0.559 ±0.024 | 0.672 ±0.032 | 0.728 ±0.022 |
| | D2D | 0.500 ±0.000 | 0.500 ±0.000 | 0.500 ±0.000 | 0.500 ±0.000 | 0.500 ±0.000 | 0.500 ±0.000 |
| | GRAPHERASER | 0.541 ±0.002 | 0.500 ±0.000 | 0.523 ±0.013 | 0.500 ±0.000 | 0.522 ±0.009 | 0.500 ±0.000 |
| | GRAPHEDITOR | 0.735 ±0.037 | 0.611 ±0.018 | - | - | - | - |
| | CERTUNLEARN | 0.721 ±0.033 | 0.602 ±0.013 | - | - | - | - |
| | GNNDELETE | **0.956 ±0.004** | **0.859 ±0.002** | **0.949 ±0.003** | **0.859 ±0.005** | **0.924 ±0.007** | **0.898 ±0.019** |

**Table 18: Unlearning task: edge deletion when $\mathcal{E}_d = \mathcal{E}_{d,\text{IN}}$. Evaluation: link prediction. Dataset: DBLP.** Reported is AUROC ($\uparrow$) performance with the best marked in **bold** and the second best underlined.

| Ratio (%) | Model | GCN $\mathcal{E}_t$ | $\mathcal{E}_d$ | GAT $\mathcal{E}_t$ | $\mathcal{E}_d$ | GIN $\mathcal{E}_t$ | $\mathcal{E}_d$ |
|---|---|---|---|---|---|---|---|
| 0.5 | RETRAIN | 0.965 ±0.003 | 0.496 ±0.028 | 0.957 ±0.002 | 0.513 ±0.021 | 0.934 ±0.005 | 0.571 ±0.035 |
| | GRADASCENT | 0.556 ±0.018 | 0.657 ±0.008 | 0.511 ±0.023 | 0.612 ±0.107 | 0.678 ±0.084 | 0.573 ±0.045 |
| | D2D | 0.500 ±0.000 | 0.500 ±0.000 | 0.500 ±0.000 | 0.500 ±0.000 | 0.500 ±0.000 | 0.500 ±0.000 |
| | GRAPHERASER | 0.515 ±0.001 | 0.500 ±0.000 | 0.523 ±0.000 | 0.500 ±0.000 | 0.507 ±0.003 | 0.500 ±0.000 |
| | GRAPHEDITOR | 0.781 ±0.026 | 0.479 ±0.017 | - | - | - | - |
| | CERTUNLEARN | 0.742 ±0.021 | 0.482 ±0.013 | - | - | - | - |
| | GNNDELETE | **0.951 ±0.002** | **0.829 ±0.006** | **0.928 ±0.004** | **0.889 ±0.011** | **0.906 ±0.009** | **0.736 ±0.012** |
| 2.5 | RETRAIN | 0.964 ±0.003 | 0.506 ±0.013 | 0.956 ±0.002 | 0.525 ±0.012 | 0.931 ±0.005 | 0.581 ±0.014 |
| | GRADASCENT | 0.555 ±0.066 | 0.594 ±0.063 | 0.501 ±0.020 | 0.592 ±0.017 | 0.700 ±0.025 | 0.524 ±0.017 |
| | D2D | 0.500 ±0.000 | 0.500 ±0.000 | 0.500 ±0.000 | 0.500 ±0.000 | 0.500 ±0.000 | 0.500 ±0.000 |
| | GRAPHERASER | 0.527 ±0.002 | 0.500 ±0.000 | 0.538 ±0.013 | 0.500 ±0.000 | 0.517 ±0.009 | 0.500 ±0.000 |
| | GRAPHEDITOR | 0.776 ±0.025 | 0.432 ±0.009 | - | - | - | - |
| | CERTUNLEARN | 0.718 ±0.032 | 0.475 ±0.011 | - | - | - | - |
| | GNNDELETE | **0.934 ±0.002** | **0.748 ±0.006** | **0.914 ±0.007** | **0.774 ±0.015** | **0.897 ±0.006** | **0.740 ±0.015** |
| 5.0 | RETRAIN | 0.963 ±0.003 | 0.504 ±0.006 | 0.955 ±0.002 | 0.528 ±0.007 | 0.931 ±0.006 | 0.578 ±0.009 |
| | GRADASCENT | 0.555 ±0.060 | 0.581 ±0.073 | 0.490 ±0.022 | 0.551 ±0.030 | 0.723 ±0.032 | 0.516 ±0.042 |
| | D2D | 0.500 ±0.000 | 0.500 ±0.000 | 0.500 ±0.000 | 0.500 ±0.000 | 0.500 ±0.000 | 0.500 ±0.000 |
| | GRAPHERASER | 0.509 ±0.011 | 0.500 ±0.000 | 0.511 ±0.006 | 0.500 ±0.000 | 0.503 ±0.000 | 0.500 ±0.000 |
| | GRAPHEDITOR | 0.736 ±0.023 | 0.430 ±0.011 | - | - | - | - |
| | CERTUNLEARN | 0.694 ±0.026 | 0.441 ±0.008 | - | - | - | - |
| | GNNDELETE | **0.917 ±0.005** | **0.713 ±0.007** | **0.912 ±0.007** | **0.733 ±0.018** | **0.864 ±0.005** | **0.732 ±0.008** |

**Table 19: Unlearning task: edge deletion when $\mathcal{E}_d = \mathcal{E}_{d,\text{OUT}}$. Evaluation: link prediction. Dataset: OGB-Collab.** Reported is AUROC ($\uparrow$) performance with the best marked in **bold** and the second best underlined.

| Ratio (%) | Model | GCN $\mathcal{E}_t$ | $\mathcal{E}_d$ | GAT $\mathcal{E}_t$ | $\mathcal{E}_d$ | GIN $\mathcal{E}_t$ | $\mathcal{E}_d$ |
|---|---|---|---|---|---|---|---|
| 0.5 | RETRAIN | 0.986 ±0.001 | 0.553 ±0.004 | 0.983 ±0.001 | 0.541 ±0.002 | 0.859 ±0.011 | 0.523 ±0.003 |
| | GRADASCENT | 0.606 ±0.012 | 0.509 ±0.002 | 0.674 ±0.016 | 0.535 ±0.020 | 0.665 ±0.097 | 0.511 ±0.009 |
| | D2D | 0.500 ±0.000 | 0.500 ±0.000 | 0.500 ±0.000 | 0.500 ±0.000 | 0.500 ±0.000 | 0.500 ±0.000 |
| | GRAPHERASER | 0.544 ±0.000 | 0.500 ±0.000 | 0.551 ±0.004 | 0.500 ±0.000 | 0.513 ±0.009 | 0.500 ±0.000 |
| | GRAPHEDITOR | 0.857 ±0.010 | 0.497 ±0.004 | - | - | - | - |
| | CERTUNLEARN | 0.841 ±0.005 | 0.478 ±0.006 | - | - | - | - |
| | GNNDELETE | **0.985 ±0.002** | **0.723 ±0.002** | **0.983 ±0.001** | **0.728 ±0.007** | **0.977 ±0.004** | **0.715 ±0.005** |
| 2.5 | RETRAIN | 0.986 ±0.001 | 0.553 ±0.001 | 0.983 ±0.001 | 0.541 ±0.002 | 0.858 ±0.006 | 0.525 ±0.002 |
| | GRADASCENT | 0.531 ±0.047 | 0.552 ±0.006 | 0.562 ±0.032 | 0.543 ±0.003 | 0.645 ±0.094 | 0.513 ±0.008 |
| | D2D | 0.500 ±0.000 | 0.500 ±0.000 | 0.500 ±0.000 | 0.500 ±0.000 | 0.500 ±0.000 | 0.500 ±0.000 |
| | GRAPHERASER | 0.517 ±0.000 | 0.500 ±0.000 | 0.524 ±0.009 | 0.500 ±0.000 | 0.542 ±0.001 | 0.500 ±0.000 |
| | GRAPHEDITOR | 0.839 ±0.002 | 0.465 ±0.002 | - | - | - | - |
| | CERTUNLEARN | 0.822 ±0.002 | 0.473 ±0.010 | - | - | - | - |
| | GNNDELETE | **0.983 ±0.002** | **0.642 ±0.002** | **0.983 ±0.000** | **0.639 ±0.003** | **0.963 ±0.009** | **0.647 ±0.007** |

**Table 20: Unlearning task: edge deletion when $\mathcal{E}_d = \mathcal{E}_{d,\text{IN}}$. Evaluation: link prediction. Dataset: OGB-Collab.** Reported is AUROC ($\uparrow$) performance with the best marked in **bold** and the second best underlined.

| Ratio (%) | Model | GCN $\mathcal{E}_t$ | $\mathcal{E}_d$ | GAT $\mathcal{E}_t$ | $\mathcal{E}_d$ | GIN $\mathcal{E}_t$ | $\mathcal{E}_d$ |
|---|---|---|---|---|---|---|---|
| 0.5 | RETRAIN | 0.986 ±0.001 | 0.521 ±0.002 | 0.983 ±0.001 | 0.644 ±0.002 | 0.855 ±0.007 | 0.599 ±0.003 |
| | GRADASCENT | 0.552 ±0.093 | 0.505 ±0.003 | 0.559 ±0.154 | 0.569 ±0.034 | 0.658 ±0.088 | 0.524 ±0.014 |
| | D2D | 0.500 ±0.000 | 0.500 ±0.000 | 0.500 ±0.000 | 0.500 ±0.000 | 0.500 ±0.000 | 0.500 ±0.000 |
| | GRAPHERASER | 0.554 ±0.002 | 0.500 ±0.000 | 0.512 ±0.003 | 0.500 ±0.000 | 0.532 ±0.002 | 0.500 ±0.000 |
| | GRAPHEDITOR | 0.840 ±0.002 | 0.571 ±0.003 | - | - | - | - |
| | CERTUNLEARN | 0.825 ±0.004 | 0.583 ±0.005 | - | - | - | - |
| | GNNDELETE | **0.976 ±0.002** | **0.715 ±0.002** | **0.974 ±0.001** | **0.814 ±0.011** | **0.935 ±0.005** | **0.663 ±0.014** |
| 2.5 | RETRAIN | 0.986 ±0.001 | 0.532 ±0.002 | 0.982 ±0.001 | 0.650 ±0.004 | 0.845 ±0.010 | 0.612 ±0.003 |
| | GRADASCENT | 0.577 ±0.005 | 0.534 ±0.006 | 0.697 ±0.040 | 0.541 ±0.018 | 0.568 ±0.097 | 0.585 ±0.004 |
| | D2D | 0.500 ±0.000 | 0.500 ±0.000 | 0.500 ±0.000 | 0.500 ±0.000 | 0.500 ±0.000 | 0.500 ±0.000 |
| | GRAPHERASER | 0.539 ±0.001 | 0.500 ±0.000 | 0.542 ±0.013 | 0.500 ±0.000 | 0.528 ±0.009 | 0.500 ±0.000 |
| | GRAPHEDITOR | 0.811 ±0.002 | 0.575 ±0.001 | - | - | - | - |
| | CERTUNLEARN | 0.803 ±0.005 | 0.547 ±0.002 | - | - | - | - |
| | GNNDELETE | **0.972 ±0.003** | **0.665 ±0.002** | **0.966 ±0.005** | **0.772 ±0.015** | **0.974 ±0.003** | **0.675 ±0.010** |

**Table 21: Unlearning task: edge deletion when $\mathcal{E}_d = \mathcal{E}_{d,\text{OUT}}$. Evaluation: link prediction. Dataset: WordNet18.** Reported is AUROC ($\uparrow$) performance with the best marked in **bold** and the second best underlined.

| c Ratio (%) | Model | R-GCN $\mathcal{E}_t$ | $\mathcal{E}_d$ | R-GAT $\mathcal{E}_t$ | $\mathcal{E}_d$ |
|---|---|---|---|---|---|
| 0.5 | RETRAIN | 0.801 ±0.007 | 0.601 ±0.014 | 0.898 ±0.003 | 0.808 ±0.015 |
| | GRADASCENT | 0.499 ±0.002 | 0.501 ±0.003 | 0.495 ±0.001 | 0.411 ±0.012 |
| | D2D | 0.500 ±0.000 | 0.500 ±0.000 | 0.500 ±0.000 | 0.500 ±0.000 |
| | GRAPHERASER | 0.517 ±0.001 | 0.511 ±0.003 | 0.533 ±0.004 | 0.508 ±0.001 |
| | GNNDELETE | **0.757 ±0.005** | **0.901 ±0.008** | **0.899 ±0.002** | **0.828 ±0.010** |
| 2.5 | RETRAIN | 0.804 ±0.005 | 0.639 ±0.004 | 0.897 ±0.004 | 0.831 ±0.005 |
| | GRADASCENT | 0.493 ±0.002 | 0.489 ±0.005 | 0.491 ±0.001 | 0.424 ±0.015 |
| | D2D | 0.500 ±0.000 | 0.500 ±0.000 | 0.500 ±0.000 | 0.500 ±0.000 |
| | GRAPHERASER | 0.515 ±0.004 | 0.508 ±0.005 | 0.533 ±0.003 | 0.505 ±0.004 |
| | GNNDELETE | **0.758 ±0.005** | **0.902 ±0.006** | **0.898 ±0.002** | **0.836 ±0.005** |
| 5.0 | RETRAIN | 0.801 ±0.004 | 0.661 ±0.011 | 0.896 ±0.001 | 0.861 ±0.006 |
| | GRADASCENT | 0.493 ±0.001 | 0.485 ±0.002 | 0.491 ±0.001 | 0.425 ±0.025 |
| | D2D | 0.500 ±0.000 | 0.500 ±0.000 | 0.500 ±0.000 | 0.500 ±0.000 |
| | GRAPHERASER | 0.525 ±0.002 | 0.504 ±0.001 | 0.531 ±0.004 | 0.502 ±0.003 |
| | GNNDELETE | **0.756 ±0.004** | **0.910 ±0.006** | **0.897 ±0.002** | **0.852 ±0.004** |

Table 22: **Unlearning task: edge deletion when $\mathcal{E}_d = \mathcal{E}_{d,\mathbf{IN}}$. Evaluation: link prediction. Dataset: WordNet18**. Reported is AUROC ($\uparrow$) performance with the best marked in **bold** and the second best underlined.

| Ratio (%) | Model | R-GCN | | R-GAT | |
|---|---|---|---|---|---|
| | | $\mathcal{E}_t$ | $\mathcal{E}_d$ | $\mathcal{E}_t$ | $\mathcal{E}_d$ |
| 0.5 | RETRAIN | 0.802 $\pm$0.007 | 0.584 $\pm$0.005 | 0.898 $\pm$0.002 | 0.771 $\pm$0.011 |
| | GRADASCENT | 0.496 $\pm$0.002 | 0.486 $\pm$0.005 | 0.493 $\pm$0.001 | 0.487 $\pm$0.008 |
| | D2D | 0.500 $\pm$0.000 | 0.500 $\pm$0.000 | 0.500 $\pm$0.000 | 0.500 $\pm$0.000 |
| | GRAPHERASER | 0.505 $\pm$0.002 | 0.510 $\pm$0.001 | 0.502 $\pm$0.000 | 0.502 $\pm$0.000 |
| | GNNDELETE | **0.756** $\pm$**0.005** | **0.850** $\pm$**0.005** | **0.897** $\pm$**0.002** | **0.819** $\pm$**0.014** |
| 2.5 | RETRAIN | 0.800 $\pm$0.005 | 0.580 $\pm$0.006 | 0.891 $\pm$0.005 | 0.783 $\pm$0.009 |
| | GRADASCENT | 0.490 $\pm$0.001 | 0.477 $\pm$0.006 | 0.490 $\pm$0.001 | 0.492 $\pm$0.003 |
| | D2D | 0.500 $\pm$0.000 | 0.500 $\pm$0.000 | 0.500 $\pm$0.000 | 0.500 $\pm$0.000 |
| | GRAPHERASER | 0.512 $\pm$0.003 | 0.509 $\pm$0.004 | 0.545 $\pm$0.015 | 0.509 $\pm$0.005 |
| | GNNDELETE | **0.751** $\pm$**0.006** | **0.845** $\pm$**0.007** | **0.893** $\pm$**0.002** | **0.786** $\pm$**0.004** |
| 5.0 | RETRAIN | 0.797 $\pm$0.003 | 0.588 $\pm$0.005 | 0.883 $\pm$0.002 | 0.786 $\pm$0.005 |
| | GRADASCENT | 0.491 $\pm$0.001 | 0.480 $\pm$0.004 | 0.490 $\pm$0.002 | 0.494 $\pm$0.002 |
| | D2D | 0.500 $\pm$0.000 | 0.500 $\pm$0.000 | 0.500 $\pm$0.000 | 0.500 $\pm$0.000 |
| | GRAPHERASER | 0.507 $\pm$0.003 | 0.511 $\pm$0.005 | 0.518 $\pm$0.002 | 0.504 $\pm$0.004 |
| | GNNDELETE | **0.749** $\pm$**0.005** | **0.850** $\pm$**0.008** | **0.889** $\pm$**0.002** | **0.779** $\pm$**0.007** |

