# OpenReview forum: "GNNDelete: A General Strategy for Unlearning in Graph Neural Networks"
_ICLR.cc/2023/Conference — ICLR 2023 poster_

### Official Review · Reviewer_ctZT · 2022-10-15

**Confidence:** 3
**Correctness:** 3
**Technical Novelty And Significance:** 3
**Empirical Novelty And Significance:** 3
**Recommendation:** 6

**Clarity, Quality, Novelty And Reproducibility:**

Clarity: The paper is largely clear. I can easily understand the main motivation and overall design of the paper. However, some points are not clear. Please refer to "Strengths and Weaknesses".

Quality: The paper is of good technical quality. The proposed method is largely sound, and there are extensive experiments, including ability to maintain performance, ability to forget knowledge, and efficiency.

Novelty: The paper is of good novelty. The authors make a valid observation that the main difference between graph data and image/text data is that deleting a node/edge directly influences other nodes and edges. Thus, dividing the input data leads to performance compromise. The proposed method, which is designed based on the observations, seem simple and effective.

Reproducibility: Code is anonymously provided. I am satisfied with it.

**Strength And Weaknesses:**

Strengths:
- This paper studies an important topic. Graph unlearning is related to privacy-preservation on graphs and machine learning.
- This paper clearly states the motivation and its difference with related works. The main difference between graph data and image/text data is that deleting a node/edge directly influences other nodes and edges. Thus, dividing the input data leads to performance compromise.
- The proposed standards and techniques are sound. The design of GNNDelete as an add-on module is interesting as it is more efficient than retraining the whole model.
- Overall, the paper is largely organized clearly and easy to follow.

Weaknesses and questions.
- Discussion with respect to dynamic network embedding is not clear. From my perspective, graph unlearning can be seen as a special case of dynamic network embedding, where the dynamics are limited to edge deletion. Thus, a simple experiment of dynamic network embedding (e.g. treating edge deletions as graph dynamics) can be added to better justify the difference.
- GNNDelete seems unable to deal with a sequence of delete requests. In the abstract, the authors state that graph unlearning is "a sequence of requests arrives to delete graph elements". However, from my understanding, it seems that GNNDelete can only handle one set of requests given at the same time. For example, suppose we are given a sequence of $T$ deletion requests, we should have $T$ separate del operators (Eqn. 3), one being responsible for each deletion request (as at earlier stages, one cannot know the requests that are yet to come). However, the authors state that "the weights $W_D^l$ are shared across all nodes" without mentioning that there can be multiple weights. Thus, I think the authors should clearly state whether and how GNNDelete handles a sequence of deletion requests (as stated in the abstract).
- There are some contradictory descriptions on the two standards. In the introduction, the authors state that "the predicted probability for deleted edges of the unlearned model should be similar to those for **nonexistent** edges". However, in Section 4.1, the authors state that "the combination of the node representations that are present in the deleted edge should be **random**". These two descriptions seem contradictory, and some justifications are needed.
- Some details and notations are not clear and can be revised. For example, in section 4.2, the Del operator (Eqn. 3), the condition $u\in S_{uv}^l$ is hard to understand as it includes two $u$.  Further, it is not described what the $\mathcal{L}$ is in Eqn. 5 and 6. Finally, I would suggest the authors clearly describe what the metrics (AUROC on $\mathcal{E}_t, \mathcal{E}_d$ and MI ratio reflect (e.g. AUROC on E_t tests ability to maintain model performance. MI-ratio and AUROC on E_d test ability of GNNDelete to delete knowledge). At the current stage, such information is not immediate without referring to the appendix.
- It seems that GNNDelete cannot perform effectively on node classification. As shown in Table 6, GNNDelete is outperformed by GraphEditor in terms of accuracy, F1 and MI-ratio. Justifying the performance difference between node classification and link prediction can strengthen this paper.

**Summary Of The Paper:**

 Graph unlearning refers to the task where a graph neural network (GNN) is required to remove the knowledge about a specific node or edge. This paper presents a method, GNNDelete, for performing the task of graph unlearning. The main motivation of this paper is that existing unlearning methods use divide-and-retrain, which leads to performance loss on graph datasets. Further, existing graph unlearning methods suffer from limited scope (e.g. limited model, limited task).

This paper proposes two standards that a graph unlearning method should satisfy, Deleted Edge Consistency and Neighborhood Influence. This paper then proposes a layer-wise, model-agnostic operator, GNNDelete to implement the two standards. Extensive experiments on link prediction show the effectiveness of GNNDelete in handling edge deletion and maintaining link prediction performance. Further, the efficiency of GNNDelete is good.

**Summary Of The Review:**

I recommend a weak accept at this stage. The studied problem is important and the proposed techniques and technically sound and make novel insights. However, the paper suffers from some unclear statements. If the authors can revise or clarify my questions, I am willing to further raise the score.

---

> ### Author Response · Authors · 2022-11-18
> **Response to Reviewer ctZT [1/3]**
>
> Thank you for the thoughtful feedback. Your comments were very helpful, and we are glad you felt the problem we focus on is important and timely and the manuscript was easy to follow.
>
> Below, we briefly respond to each of your concerns. If, after reading our responses, you are still not convinced of the merits of our work, we would love to know what else we can do to help improve the work towards a better score in your eyes.
>
> **RE Weakness 1:**
>
> Thank you for the insightful comments. Our response has two parts.
>
> First, following your suggestion, we conduct a new experiment comparing GNNDelete (ours) to JODIE, which is a dynamic network embedding (DNE) model. The table below shows the results of deleting 2.5% of edges on the DBLP dataset, where JODIE interprets the edge deletion as graph dynamics. We can see that **GNNDelete outperforms JODIE (DNE) by 13.1%** in predictive performance (link prediction) on the test set and by 13.5% in deletion performance (distinguishing deleted from present edges). Our experiments highlight a critical difference between Graph Unlearning and Dynamic Network Embedding, wherein dynamic networks constantly evolve while unlearning requests are usually a handful of small batches.
>
> **Deletion task: 2.5% edge deletion on DBLP. Evaluated downstream task: link prediction. Metric: AUC**
> |                                       | AUC on $\mathcal{E}_t$ | AUC on $\mathcal{E}_d$ |
> |---------------------------------------|------------------------|------------------------|
> | Retrain from scratch (reference only) | 0.964 ±0.003           | 0.506 ±0.013           |
> | JODIE (Dynamic Network Embedding)| 0.801 ±0.042           | 0.613 ±0.026           |
> | GNNDelete (Ours)                      | 0.934 ±0.002           | 0.748 ±0.006           |
>
> Second, dynamic network embedding and graph unlearning are fundamentally different problems. To explain that, we consider two use cases:
> * Nodes/edges/node features in the graph become invalid at some point and must be removed, i.e., deleted, from a trained GNN model. Dynamic network methods can be considered in that case by representing the time of a deletion request as a time stamp upon which a node/edge/node feature gets removed from the graph. While dynamic network methods are often used when we expect major changes in the connectivity of the underlying graph, graph unlearning methods are meant to be used for mostly stable graphs with a relatively small portion of data points being deletion. This is a use case that is shared between graph unlearning and dynamic network methods.
>
> * There exist some use cases that dynamic network methods cannot solve. A prominent example is a situation when a data point must be deleted because of its sensitive nature or privacy concerns (in contrast to the above-mentioned use case, the data point still represents valid information). Graph unlearning methods can handle such settings, whereas dynamic network methods cannot.
>
>
>
> **RE Weakness 2:**
>
> We design the deletion operator in a way that it can handle a sequence of deletion requests. We would like to bring to the reviewers’ notice that, in stark contrast to existing deletion techniques, we can continue training the same deletion operators without having to train separate ones or retrain from scratch.
>
> To prove that, we designed the sequence deletion experiment, where GNNDelete is asked to delete 5 batches of 0.5% of edges (a total of 2.5%), one batch after another on the DBLP dataset.
>
> **Deletion task: 2.5% sequential edge deletion, 0.5% a batch.**
> | Ratio (%) | AUC on $\mathcal{E}_t$ | AUC on $\mathcal{E}_d$ |
> |-----------|------------------------|------------------------|
> | 0.5       | 0.951                  | 0.829                  |
> | 1.0       | 0.949                  | 0.808                  |
> | 1.5       | 0.943                  | 0.791                  |
> | 2.0       | 0.938                  | 0.776                  |
> | 2.5       | 0.934                  | 0.748                  |
>
>
> **RE Weakness 3:**
>
> Sorry for the confusion and thank you for raising this point. We have made it clear in our revision. In the introduction, we now claim “predicted probability for deleted edges to be similar to those for **nonexistent** and **existent** edges”. Now, it is not contradictory to “ node representations that are present in the deleted edge should be **random**”. Random nodes are the best option when our knowledge of the input graph is incomplete (i.e., a black-box setting). Our results (Table 1-2, Response to Reviewer PAk1) show that GNNDelete successfully deletes an edge $e_{uv}$ if it minimizes the difference between node-pair representations $\phi(h_{u}, h_{v})$ and any two randomly chosen nodes from the set of all nodes in the graph.
>
> *Continuation of the response in the thread.*

---

> > ### Author Response · Authors · 2022-11-18
> > **Response to Reviewer ctZT [2/3]**
> >
> > **RE Weakness 4:**
> > We thank the reviewer for bringing up these clarity issues. We fixed the notation in the revised manuscript, and below we make an effort for a direct clarification of the issues.
> > ### Regarding Equation 3
> > We apologize for the confusion and would like to clarify. When we want to delete an edge $e_{uv}$, $S^l_{uv}$ refers to the $l$-th hop local enclosing subgraph of $e_{uv}$, with $u$ being some other node in $S^l_{uv}$ (thus the notation is overloaded in Eq. 3, which we corrected). In the revised manuscript, we replace $u$ with the letter $w$ to clarify that point such that it will be $w \in S^l_{uv}$.
> >
> > Further, the deletion operator is applied to each node embedding vector, acting as a mask, depending on whether the node belongs to the specific enclosing subgraph. In particular, when a node is out of the enclosing subgraph, the deletion operator does not affect the node embedding at all. In contrast, when a node lies in the enclosing subgraph, function $\phi$ is applied to the node embedding vector. Thanks for bringing up this point, which we clarify it in the revised manuscript.
> >
> >
> > ### Regarding Equations 5 and 6
> > We apologize for the missing information. $\mathcal{L}$ in Eq. 5 and 6 refer to an arbitrary choice of a distance function. We use Mean Squared Error (MSE) as the distance function throughout the experiments.
> >
> >
> > ### Regarding the evaluation metrics
> > (also updated in Section 5.1 and Appendix C)
> >
> > $\mathcal{E}_t$ represents the edge test dataset, that consists of existent edges in the original dataset. In order to demonstrate the robustness of our method, we compare performance on both $\mathcal{E}_t$ and $\mathcal{E}_d$. We made it clear in our revision. Below, we give a more elaborate description of the metrics, aligned with Appendix C:
> > 1. AUC on $\mathcal{E}_t$: This is a standard metric the same as in other papers that evaluate the performance of GNN predictors (not deletion models). It is used to test the model’s ability to link prediction (shown in Tables 1-2).
> > 2. AUC on $E_d$: This metric is used to test a model’s ability to distinguish remaining edges ($E_r$) from deleted edges ($E_d$) (Shown in Table 1-2). The idea is that the remaining edges ($E_r)$ should have high predicted probabilities, while the deleted edges ($E_d$) should have low predicted probabilities, i.e. $g(E_d) < g(E_r)$, where g is the link prediction decoder. The higher the AUC on $E_d$, the better the model can distinguish $E_d$ from $E_r$.
> >
> > 3. MI Ratio: It is used to quantify privacy leakage (Shown in Table 3) For its definition, let MI$(\cdot)$ denote the membership inference model. Then, we have the following steps:
> >     * Get the existence probability of Ed before deletion, MI$_\text{before}$$(E_d)$
> >     * Get the existence probability of Ed after deletion, MI$_\text{after}$$(E_d)$
> >     * Compute the before-to-after ratio of existence probability, MI ratio $=  \frac{\text{MI}_\text{before}(E_d)}{\text{MI}_\text{after}(E_d)}$
> > If the ratio is higher than 1, it means that the model contains less information about Ed, thus has forgotten Ed. If the ratio is less than 1, it means the model contains more information about Ed. (quantified by the MI attack model)
> >
> > *Continuation of the response in the thread.*

---

> > > ### Author Response · Authors · 2022-11-18
> > > **Response to Reviewer ctZT [3/3]**
> > >
> > > **RE Weakness 5:**
> > >
> > > Thank you for the great point on node classification performance. In response to your comment, we repeated our node deletion experiments on the DBLP dataset and did a complete hyperparameter search. These new results show that GNNDelete outperforms baselines on node classification performance while deleting _nodes_. GNNDelete outperforms GraphEditor by 1.8% accuracy and 1.9% F1 score. It is also 1.4% better than GraphEditor in terms of membership inference attacks.
> > >
> > > **Deletion task: 100 node deletion on DBLP. Evaluated downstream task: node classification.**
> > > |                                       | Acc          | F1           | MI Ratio     |
> > > |---------------------------------------|--------------|--------------|--------------|
> > > | Retrain from scratch (reference only) | 0.845 ±0.008 | 0.841 ±0.004 | 1.515 ±0.034 |
> > > | Gradient ascent                                  | 0.392 ±0.026 | 0.341 ±0.035 | 1.021 ±0.113 |
> > > | D2D                                                    | 0.250 ±0.000 | 0.250 ±0.000 | **1.755 ±0.065** |
> > > | GraphEraser                                       | 0.718 ±0.014 | 0.716 ±0.011 | 0.975 ±0.083 |
> > > | GraphEditor                                        | _0.765 ±0.012_ | _0.749 ±0.006_ | 1.260 ±0.088 |
> > > | GNNDelete (Ours)                              | **0.793 ±0.016** | **0.768 ±0.009** | _1.401 ±0.082_ |
> > >
> > > We also show that GNNDelete can outperform baselines on node classification tasks while deleting _edges_ as well.
> > >
> > > **Deletion task: 2.5% edge deletion on DBLP. Shown is the performance on the downstream task: node classification. Metric: Accuracy**
> > > |                                       |    Acc on GCN     |    Acc on GAT     |    Acc on GIN     |
> > > |---------------------------------------|:-----------:|:-----------:|:-----------:|
> > > | Retrain from scratch (reference only) | 0.821±0.012 | 0.830±0.012 | 0.817±0.012 |
> > > | Gradient ascent                       | 0.532±0.055 | 0.547±0.024 | 0.550±0.041 |
> > > | D2D                                   | 0.519±0.008 | 0.511±0.004 | 0.513±0.007 |
> > > | GraphEraser                           | 0.681±0.037 | 0.694±0.022 | 0.673±0.033 |
> > > | GraphEditor                           | 0.667±0.024 | 0.673±0.019 | 0.680±0.031 |
> > > | Ours                                  | **0.802±0.018** | **0.799±0.015** | **0.812±0.017** |
> > >
> > >
> > > Even though GNNDelete only outperforms GraphEditor on node classification (when deleting nodes) by a small margin, we identify three advantages of GNNDelete over GraphEditor:
> > > 1. GNNDelete is broadly applicable to any downstream tasks, including node classification, link prediction, and graph classification. In contrast, GraphEditor does not apply to link prediction and graph classification as downstream tasks (see Responses to Reviewer PAk1).
> > > 2. GNNDelete is model-agnostic and can be integrated with any existing trained GNN model. While GraphEditor requires extra efforts to work on non-linear GNNs and requires caching the closed-form solution when training the underlying GNN predictor, which significantly limits its usability.
> > > 3. GNNDelete takes less computational cost, both space-wise and time-wise, than GraphEditor. GraphEditor takes 18.5% more time to train than GNNDelete, for a 2.5% edge deletion task on DBLP, including finding closed-form solutions and additional finetuning.
> > >
> > > **Conclusion**
> > >
> > > We thank you very much again for your insightful feedback! We would greatly appreciate knowing if we can make any other revisions and offer additional clarification to address your concerns. We look forward to hearing from you–thank you!

---

> > > > ### Comment · Reviewer_ctZT · 2022-11-18
> > > > **Your detailed response is well received.**
> > > >
> > > > Dear authors,
> > > >
> > > > I have read your response. I acknowledge your clarifications on:
> > > > - Difference between dynamic network embedding and graph unlearning. The added experiments further underscore the need to develop graph unlearning techniques.
> > > > - Whether the deleted node pairs should resemble random or non-existent edges.
> > > > - Notations and evaluation metrics.
> > > > - Implications on node classification experiments.
> > > >
> > > > I have one quick follow-up question. In the process of GNNDelete to handle a stream of deletions, is it true that,
> > > > - if two edges on a node are deleted at different timesteps, we modify the same $Del$ module parameter, instead of adding new ones?
> > > > - if an edge on a node is required to be deleted at later timesteps (instead of at first), we simply replace the $Del$ module from identity to a learnable matrix and train it?
> > > > If yes, could you please add a small paragraph to clarify that, as "handling a sequence of requests" is explicitly stated.
> > > >
> > > > Other than that, I think the response is satisfactory. I will take time to read other reviews and see whether I have follow-up questions.

---

> > > > > ### Author Response · Authors · 2022-11-21
> > > > > **Re: Response to follow up questions**
> > > > >
> > > > > Thank you for your fast response and acknowledging our response to your comments! We really appreciate your feedback.
> > > > >
> > > > > &nbsp;
> > > > > &nbsp;
> > > > > -----------
> > > > >
> > > > > **Re follow-up Q1: if two edges on a node are deleted at different timesteps, we modify the same $Del$ module parameter, instead of adding new ones?**
> > > > > Yes, that is correct. For different edges deleted at different timesteps (even if the deleted edges touch the same node), we modify the same $Del$ module parameter. No new $Del$ modules are added, which makes the approach efficient as new deletion requests arrive.
> > > > >
> > > > > **Re follow-up Q2: if an edge on a node is required to be deleted at later timesteps (instead of at first), we simply replace the $Del$ module from identity to a learnable matrix and train it**
> > > > > Yes, that is correct–GNNDelete switches the $Del$ module from an identity to a learnable matrix and train its local version based on $S_{uv}$ of the edge to be deleted. We can easily change which node representations we want to feed into the $Del$ module.
> > > > >
> > > > > **Re follow-up Q3:  If yes, could you please add a small paragraph to clarify that, as "handling a sequence of requests" is explicitly stated.**
> > > > > Thank you for bringing this up! We first define the sequential graph unlearning problem, and briefly describe how GNNDelete handles sequential unlearning. We just added the following paragraph to the revised version.
> > > > >
> > > > > Problem Formulation: Sequential Graph Unlearning
> > > > > Given a graph $G = (\mathcal V, \mathcal{E}, \mathbf{X})$, a fully trained GNN model $m(G)$, and a sequence of edges $\mathcal E_{seq,d} = \{ \mathcal E_{d,0}, \mathcal E_{d,1}, …, \mathcal E_{d,S} \}, |\mathcal E_{seq,d}| = S$,  where each deletion request $\mathcal E_{d,i} \in \mathcal E_{seq,d}$ is a standard graph unlearning problem (as defined in Sec. 4).  Then, sequential graph unlearning aims to unlearn every batch of edges $\mathcal E_{d,i} \in \mathcal E_{seq,d}$ from the GNN model $m(G)$ in a **sequential** manner meaning that a request $\mathcal E_{d,i+1}$ is given to the unlearned model after $\mathcal E_{d,i}$ has been successfully unlearned.
> > > > >
> > > > >
> > > > > GNNDelete is designed such that it can handle a sequence of deletion requests. In stark contrast to existing deletion techniques (GraphEraser, GraphEditor, Descent-to-Delete), GNNDelete can continue training the same deletion operator **without** having to train a separate one for each individual deletion request or retrain the deletion operator from scratch for each new request. As $\mathcal E_{d,i} \neq \mathcal E_{d,i+1}$, we can maintain a binary mask to easily turn on and off the deletion operator for specific nodes following the definition in Eq.3. This is specified by the binary mask containing information about what node representations are to be updated in the $Del$ operator.
> > > > >
> > > > > &nbsp;
> > > > > &nbsp;
> > > > > &nbsp;
> > > > > &nbsp;
> > > > > -----------
> > > > > We hope our updated manuscript and responses have addressed your comments. We would greatly appreciate your acknowledging our response and updating the scores if satisfied. If any concerns remain, we would be happy to respond. Thank you!

---

### Official Review · Reviewer_Wk6L · 2022-10-25

**Confidence:** 4
**Clarity, Quality, Novelty And Reproducibility:** It is understandable to a large exten…
**Correctness:** 2
**Technical Novelty And Significance:** 2
**Empirical Novelty And Significance:** 3
**Recommendation:** 3

**Strength And Weaknesses:**

Strength

1. The proposed method does not require extra information from the training process, which is required in many unlearning schemes.

2. The proposed method can also be used for deleting nodes by removing all their associated edges.

3. The algorithm is evaluated on a number o freal graph  datasets to demonstrate the effectiveness of the proposed approach for machine unlearning.

Weakness

1. It is hard to perceive the paper's motivation of minimizing the change in the neighborhoods of deleted edges. If one expects the information of an edge to be removed entirely from the trained model, its neighborhood's intermediate representation must be changed to reflect the removal. Otherwise, its influence will most probably remain in the network.

2. The predicted probability of a deleted edge should not be random, as enforced in the paper. Instead, it should match the posterior distribution of the edge existing, given the corresponding node information. Otherwise, these deleted edges can be potentially caught by abnormal/outlier detectors and thus be recovered from deletion.

3. The proposed removal process seems to be data independently reversible, which is the last thing to hope for from a data provider’s aspect. Suppose the original untouched network can be recovered by removing the patching network described in the paper. How would you be able to provide convincing evidence to the data providers that their data’s influence has been eliminated once and for good?

4. Theorem 1 only shows that the predicted probability for the target edge of deletion will change for at least a certain amount but tells nothing about its influence being eliminated. There is also seemingly no effort that can be found in the paper to regularize the behavior of the unlearned model and a clean, trained-from-scratch model with target edges removed from its training dataset.

**Summary Of The Paper:**

The paper proposed an unlearning method for GNN. In order to remove an edge, a patch network is attached to each layer of the original network, which are MLPs that only affects nodes in the neighborhood of the target edges. These networks are trained so that the neighborhood representations remain the same while that of the two nodes of the target edge is tuned to be similar to random pairs of vertices sampled from the graph.

**Summary Of The Review:**

In general, the studied problem is interesting and important. In addition, the methodology is principled with three major merits as discussed above. However, the work still has some unaddressed concerns to well justify its technical contributions.

---

> ### Author Response · Authors · 2022-11-18
> **Response to Reviewer Wk6L [1/3]**
>
> Thank you very much for your valuable perspective! Below, we respond in detail to each of your concerns; if, after reading our response, you do not feel we have sufficiently justified a higher score, please let us know where we can further improve our work or what concerns you still have so we can improve the work further. Thank you again! We look forward to hearing from you.
>
> **Re Weakness 1: "... It is hard to perceive the paper's motivation for minimizing the change in the neighborhoods of deleted edges."**
>
> We apologize for the confusion regarding our motivation and are happy to clarify this paper’s motivation. While we agree with the reviewer’s comment that one should change the representation of the edge’s entire neighborhood for efficient edge unlearning, we need to also ensure that unlearning does not lead to a drop in predictive performance involving nodes, edges, etc., that were not in the deletion requests. That is, the predictive performance of the unlearned GNN model should also be evaluated on the whole remaining graph to confirm that the prediction performance is retained for unaffected nodes.
>
> That is exactly why we introduce a novel **Neighborhood Influence (NI)** property that leverages Granger causality to remove information about edge $e_{uv}$ from enclosing subgraphs around nodes $u$ and $v$ while ensuring that remaining edges in subgraph $S_{uv}$ get high-quality representations that are necessary for accurate prediction. Our extensive empirical analysis (see results in Table 1-2 and also our response to the 3rd question of Reviewer PAk1) show that GNNDelete achieves strong deletion performance for deleted edges $E_{d}$ and retains the predictive performance of GNNs on the remaining edges $E_{r}$. We are very grateful to the reviewer for recognizing the critical role in changing $e_{uv}$’s neighborhood intermediate representations to reflect the deletion.
>
> **Re Weakness 2: "... The predicted probability of a deleted edge should not be random ..."**
>
> We would like to clarify a potential misunderstanding here and have changed the statement in the revised manuscript. We do not use a random distribution while optimizing for the **Deleted Edge Consistency** property. We say that GNNDelete successfully deletes an edge $e_{uv}$ if it minimizes the difference between node-pair representations $\phi(h_{u}, h_{v})$ and any two randomly chosen nodes from the set of all nodes in the graph. In addition, we take the expectation from a set of multiple pairs of random nodes, which may include nodes with both existing and non-existing edges, so that the resulting representations are not biased towards a specific set of node pairs. The choice of random nodes is the best option when our knowledge of the input graph is incomplete (i.e., a black-box setting), **which is our graph unlearning setup and the main focus of our work.**
> In addition, we also performed additional experiments to verify if we could detect the deleted edges using outlier detectors. We trained an Elliptic Envelope model to detect deleted edges on the DBLP dataset and compute how many edges are considered non-outliers. Our results below and in Section C of the revised Appendix show that after unlearning, GNNDelete can better prevent the deleted edges from being detected by outlier detectors. GNNDelete can protect 7% more edges than GraphEditor and 15% more than GraphEraser.
>
> **Deletion task: 2.5% edge deletion on DBLP. Evaluated task: outlier detection.**
> |                                       | Percentage of edges classified as non-outliers (higher is better) |
> |---------------------------------------|-------------------------------------------|
> | Retrain from scratch (reference only) | 0.746                                     |
> | Gradient ascent                       | 0.503                                     |
> | D2D                                   | 0.517                                     |
> | GraphEraser                           | 0.563                                     |
> | GraphEditor                           | 0.642                                     |
> | GNNDelete (Ours)                      | 0.710                                     |
>
> *Continuation of the response in the thread.*

---

> > ### Author Response · Authors · 2022-11-18
> > **Response to Reviewer Wk6L [2/3]**
> >
> > *Continuation of the response.*
> >
> > **Re Weakness 3: " ... Can we recover information by removing the patching network? ..."**
> >
> > Thank you for the insightful question. Unlike other data modalities (e.g., images and texts), unlearning for models trained on graph-structured data is fundamentally different because of dependencies between nodes connected by edges. We agree with the reviewer that attackers can trivially recover the edge information from its neighboring nodes. However, this is exactly why we propose the **Deleted Edge Consistency (DIC)** and **Neighborhood Influence (NI)** properties (described in Section 4.1) for graph unlearning, which prevents the recovery of deleted edges from their local neighborhood. While DIC reduces the predicted probabilities from unlearned models for deleted edges, NI removes the influence of the deleted edges from their respective subgraphs. This is also corroborated by our empirical results using Membership Inference (MI) Attacks (Section 5.3), which show that across five GNN architectures, GNNDelete improves the MI performance across the majority of existing baselines.
> >
> > **"... How would you be able to provide convincing evidence to the data providers that their data’s influence has been eliminated once and for good? ..."**
> >
> > This is a great question that has far-reaching implications for the progress of the field! We would like to bring to the reviewer’s notice that graph unlearning is a nascent subfield with a very little theoretical grounding, and there is no work on benchmarking the performance of graph unlearning techniques either.
> >
> > To start addressing this critical gap, we follow previous studies [1] and use standard performance metrics, such as F-1 score, accuracy, and AUROC to evaluate the predictive performance of unlearned models. In addition, we quantify the privacy leakage of deleted nodes/edges/node features using Membership Inference (MI) attacks, which is a robust metric to determine whether a given data point has been used in a model's training process. Intuitively, if a data point is successfully unlearned, then an MI attacker should not be able to infer that data point from the unlearned GNN model.
> >
> > [1] Chen et al. Graph Unlearning. ACM SIGSAC Conference on Computer and Communications Security 2022.
> >
> > *Continuation of the response in the thread.*

---

> > > ### Author Response · Authors · 2022-11-18
> > > **Response to Reviewer Wk6L [3/3]**
> > >
> > > **Re Weakness 4: "... Theorem 1 only shows ... training dataset ..."**
> > >
> > > We apologize for the lack of details. We would like to clarify that the primary aim of our theoretical analysis is to present a lower bound that estimates the minimum cost (or embedding difference) a GNNDelete operation must incur to unlearn a given edge $e_{uv}$ successfully. Consequently, we do not expect a significant change in the output representations as that would affect the predictive performance of the undeleted edges. That’s an important finding because, to the best of our knowledge, our study is the first work proposing a **layer-wise unlearning** operator. Ensuring that its output representations are not affected by a radical change makes our DEL operator a **safe-to-use** operator that is able to be incorporated into various models.
> > >
> > > Regarding the connection with the influence of the neighbors, Theorem 1 can give insights into how neighborhood representations are impacted. That is because the message-passing step will consider the representations of the nodes that are part of a deleted edge $e_{uv}$ as bounded by the bound term in Equation 4. For example, let’s consider a pair of nodes $u’,v’$ that are not being deleted but are rather a part of $e_{uv}$’s enclosing subgraph. Then, according to Equation 8 (see Appendix B), their node representations will be changed only by the neighborhood terms that are also part of the enclosing subgraph of $e_{uv}$, where the latter ones are bounded according to Equation 4, while the rest of the node representations should remain the same. That being said, although Theorem 1 is about the nodes of a deleted edge, it gives insights into the influence over the neighboring nodes.
> > >
> > > We would like to bring to the reviewer’s notice that we compared the performance behavior of a clean, trained-from-scratch model with GNNDelete in all graph unlearning experiments (Sections 5.2 and 5.3). More specifically, our results in Section 5.1 show that GNNDelete outperforms trained-from-scratch models by **21.7%**, proving its capability of effectively unlearning the deleted edges. We would be glad if the reviewer could clarify if we missed any specific concern raised by the reviewer regarding the clean, trained-from-scratch model.
> > >
> > > **Conclusion**
> > > We are very grateful to the reviewer for all their suggestions. These have really helped us in improving the paper significantly. We tried our best to incorporate all the reviewer suggestions in our write-up. In light of these updates, we kindly request the reviewer to consider increasing their score. We are very happy to answer any further questions.

---

> ### Author Response · Authors · 2022-11-23
> **Looking forward to your feedback**
>
> Dear reviewer,
>
> Thank you again for your valuable suggestions on improving this manuscript. To address your concerns, we performed a number of additional experiments, included new tables and figures in the paper, and revised the text.
>
> We would love to know what you think about our response and if there is anything else we can do to improve the paper. We would greatly appreciate your considering increasing the score. Many thanks!

---

> ### Author Response · Authors · 2022-11-29
> **Looking forward to your response**
>
> Dear reviewer,
>
> Thank you again for your thoughtful commentary. Following your suggestions, we included new results and clarified notation in the initial submission. We would love to hear your thoughts on our response. Please let us know if there is anything else we can do to address your comments. We would be very grateful if you considered increasing the score.

---

> ### Author Response · Authors · 2022-12-08
> **Looking forward to your response**
>
> Dear reviewer,
>
> We sincerely appreciate your valuable comments on our work. In our previous response and the updated manuscript, we have tried our best to address the points raised in your review. Is there any unclear point that we can further clarify? We would be very grateful if you considered increasing the score.
>
> Thank you again!

---

> ### Author Response · Authors · 2022-12-12
> **Looking forward to your feedback**
>
> Dear reviewers,
>
> Thank you again for your precious time and valuable comments! Following your suggestions, we included new results, clarifications of formulations, and theoretical analysis. We would love to hear your thoughts on our responses. Please let us know if there is anything else we can do to address your comments. We would be very grateful if you considered increasing the score.
>
> Thank you very much!

---

### Official Review · Reviewer_kuQk · 2022-11-03

**Confidence:** 3
**Correctness:** 3
**Technical Novelty And Significance:** 3
**Empirical Novelty And Significance:** 3
**Recommendation:** 6

**Clarity, Quality, Novelty And Reproducibility:**

- Clarify: This paper is written well and easy to understand.
- Quality: The quality is good. The desired properties for graph unlearning work effectively as the goals of the proposed approach.
- Novelty: I'm not very aware of previous works on graph unlearning, but I think the proposed approach has novelty.
- Reproducibility: The authors provide the code for reproducibility.



**Strength And Weaknesses:**

Strengths
1. The two desired properties for graph unlearning effectively work as the goals of the problem.
2. The proposed DEL operator can be generally used for various types of graph models.
3. The performance of the proposed approach is significantly better than those of existing approaches.

Weaknesses
1. This work studies only link prediction, while there are other tasks such as node classification or graph classification for which GNNs are widely used. I think the proposed approach can also be applied to such tasks, since the node representations can still be used to compose an edge probability even though the model is trained for a different task.
2. I’m not sure whether adding a new weight matrix to an existing model fits the purpose of “unlearning,” since the learned knowledge in the initial model does not change. If one can recover the knowledge of the original model by removing the added weights, isn’t it problematic in terms of privacy?
3. Eq. (1) forces the probability of a deleted edge to be close to the mean probability of all edges. However, even though e_{u, v} is perfectly deleted, i.e., if we re-train a GNN without e_{u, v}, the probability for e_{u, v} still have a variance and can be different from the average of all probabilities. I think this condition is too strong, and it will make more sense if we consider the probabilities collectively such as by comparing the mean of all deleted edges and that of all other edges. Can you support Eq. (1) empirically or theoretically?

Questions
1. What happens if we need to remove multiple edges? Should we run the algorithm multiple times? How long does it take?
2. In Theorem 2, why is it beneficial to bound the difference of the norm difference? I think the purpose of deletion is to change the edge prediction probability for e_{u, v}, not preserving it, especially when the readout function \phi is the dot product.

Minor comments
1. I think “Problem Formulation” in Section 4 should be clearer, since it is the formulation, not a description. For example, I suggest to replace the expressions like “the information of all edges” or “fails to predict”, which are not formally defined.


**Summary Of The Paper:**

This work solves the problem of graph unlearning, where a sequence of requests arrive to delete graph elements (nodes, edges) from trained graph neural networks (GNN). To unlearn information from a trained GNN, its influence on model weights must be deleted from the model. This work formalizes required properties for graph unlearning in the form of Deleted Edge Consistency and Neighborhood Influence. Then, it proposes GNNDelete, a model-agnostic layer-wise operator that optimizes both properties for unlearning tasks.

**Summary Of The Review:**

This paper solves the graph unlearning problem by presenting two desired properties and designing a novel approach that satisfies the two properties. I think this paper proposes a reasonable approach for the problem with strong empirical performance, but I have a few concerns  on the generalizability and the motivations of the proposed approach. Please see the weaknesses.

---

> ### Author Response · Authors · 2022-11-15
> **Response to Reviewer kuQk [1/2]**
>
> We thank you for the constructive comments! Addressing your remarks and questions can help us make our work more clear and solid. Below, we first discuss the mentioned weaknesses and then we address the two posed questions.
>
> **Re Weakness 1:**
>
> As you correctly mention, the output node representations of GNNDelete allow us for an evaluation over other tasks, such as node and graph classification. We chose to focus on the link prediction task to highlight the ability of our method to recover edge information, despite the information loss from the deleted edges.
> Based on your comments, we run additional node and graph classification experiments using GCN as the underlying predictor model and evaluated performance of GNNDelete as well as baseline methods. We show results below. **Bold** numbers indicate the best performance. *Italic* numbers indicate second best performance.
>
> | Model                                 | Acc on node classification | AUC on graph classification |
> |---------------------------------------|----------------------------|-----------------------------|
> | Retrain from scratch (reference only) | 0.821±0.012                | 0.758 ±0.068                |
> | Gradient ascent                       | 0.532±0.055                | 0.601 ±0.063                |
> | D2D                                   | 0.519±0.008                | 0.572 ±0.013                |
> | GraphEraser                           | _0.681±0.037_              | 0.596 ±0.032                |
> | GraphEditor                           | 0.667±0.024                | _0.613 ±0.035_              |
> | GNNDelete (Ours)                      | **0.802±0.018**            | **0.710 ±0.041**            |
>
> These new results show that GNNDelete can be applied to node classification and graph classification tasks. It outperforms existing unlearning methods, for example 11% by GraphEraser in node classification task and 10% by GraphEditor in graph classification task.
>
>
> **Re Weakness 2:**
>
> Let us first elaborate on how GNNDelete’s training process works. Let us have an *initial* model $m$ and the initial parameter set $\Theta_m$. For the forward step, we apply the deletion operator $\text{Del}$ to the node representations with its own parameter set $W_D$.  For the backward step, we update *only* the parameters $W_D$, keeping fixed the initial parameter set $\Theta_m$ (you can check the equations for the backward pass in Section 4.3). After $T$ epochs, the output node representations of the new *unlearned* model are affected by the learned weight parameters $W_D$.
>
> Given that the two properties are satisfied, GNNDelete’s impact is **twofold**: a) the representations for the deleted edges do not provide any information, whether they were previously existent in the graphs (Deleted Edge Consistency), and b) the representations, and hence the labels of the locally neighboring nodes are not significantly affected by the edge deletion (Neighborhood Influence). Thus, provided the unlearned model, the updated representations (affected by the GNNDelete’s parameters $W_D$) do not allow for recovering the existence information of the deleted edges, yielding efficient unlearning.
>
> **Re Weakness 3:**
>
> We thank you for your thoughtful comment. Equation 1 forces the probability of a deleted edge to be close to random, as the second term in the expectation is considered between a pair of any two nodes in the graph (connected or disconnected).
>
> Theoretically, the Deleted Edge Consistency (DEC) property aligns with the motivation of the Membership Inference (MI) attack models (Section 6.6 of [1]). The **unlearned** graph model shouldn’t be able to predict the existence or absence of a deleted edge so that there is no information leakage from the initial model.  In the case where we consider the distribution of the deleted vs the remaining edges, then we introduce a bias into our model that can bring information leakage (i.e. an MI attacker could extract information from the distribution of the deleted vs remaining edges).
>
> Empirically, the effectiveness of the DEC property is shown in Tables 1, and 2 and more specifically in the evaluation results about the AUC on the $E_d$ dataset. As we describe in Appendix C.1, the AUC on the $E_d$ measures the ability of the unlearned models to discriminate the deleted edges from the remaining edges (higher values show that the model is more capable of distinguishing edges in  $\mathcal{E}_d$ from edges in  $\mathcal{E}_r$). Thus, the higher values that GNNDelete achieves show that the **unlearned** model does not treat the deleted edges as part of the graph. Combining this observation with the results on the MI ratio (in Table 3) indicates that pushing the representations of the deleted edges to random (according to the DEC property) yields effective unlearning.
>
> [1] Graph Unlearning. Chen et al. ACM SIGSAC Conference on Computer and Communications Security 2022.
>
> *Continuation of the response in the thread*.

---

> > ### Author Response · Authors · 2022-11-15
> > **Response to Reviewer kuQk [2/2]**
> >
> > Continuation of the response.
> >
> > **Re Question 1:**
> >
> > Although the definition of the desired properties (Definitions 1, and 2) are based on a single edge deletion, in fact, we consider multiple edge requests at once during the model design.This can be seen from the definition of the loss functions (Equations 5, and 6), where we take into account the whole set $E_d$ of deleted edges. As we mention in Section 4.3, GNNdelete easily scales to larger graphs and larger numbers of edge/node deletion requests. We empirically show the time and space efficiency in Section 5.3 and Figure 2 with respect to an increasing graph size.
> >
> >  **Re Question 2:**
> >
> > The primary aim of Theorem 1 is to present a lower bound providing an estimate of the minimum cost (or embedding difference) a GNNDelete operation must incur to effectively unlearn an edge $e_{uv}$. Thus, we do not expect a significant change in the output representations as that would affect the predictive performance of the undeleted edges. This observation leads to the conclusion that the deletion operator is a safe-to-use layer-wise operator, that yields unlearned node representations of not large magnitude, that could affect the model training.
> >
> > **Re Minor comment:**
> >
> > As you correctly note, we update the problem formulation in the revised manuscript (see introductory part of Section 4) to make it more clear.
> >
> > [1] Graph Unlearning. Chen et al. ACM SIGSAC Conference on Computer and Communications Security 2022.
> >
> >
> > **Conclusion**
> >
> > We thank you very much again for your fruitful feedback! We would like to invite you to further discussion, in case your concerns are still not addressed.

---

> > > ### Comment · Reviewer_kuQk · 2022-12-07
> > > **Thank you for the Response**
> > >
> > > Dear authors,
> > >
> > > Thank you for the clarification and new experiment. I was a bit confused about the theoretical claims of the paper at first, but the concerns are addressed in your response. I also appreciate your new experiment, which I believe is important to demonstrate the generality of the proposed approach. I'm happy to increase my score.

---

> > > > ### Author Response · Authors · 2022-12-08
> > > > **Thank you for your reply!**
> > > >
> > > > Dear Reviewer,
> > > >
> > > > Thank you very much for taking the time to review our response! We are glad that our experiments and clarifications have addressed your concerns. We really appreciate your feedback and acknowledgement. And thank you for increasing the score!
> > > >
> > > > Thank you again!

---

> ### Author Response · Authors · 2022-11-23
> **Looking forward to your response**
>
> Dear reviewer,
>
> Thank you again for your thoughtful commentary. Following your suggestions, we included new results and clarified notation in the initial submission. We would love to hear your thoughts on our response. Please let us know if there is anything else we can do to address your comments. We would be very grateful if you considered increasing the score.

---

> ### Author Response · Authors · 2022-11-29
> **Looking forward to your feedback**
>
> Thank you again for your valuable suggestions on improving this manuscript. To address your concerns, we performed a number of additional experiments, included new tables and figures in the paper, and revised the text.
>
> We would love to know what you think about our response and if there is anything else we can do to improve the paper. We would greatly appreciate your considering increasing the score. Many thanks!

---

### Official Review · Reviewer_mYMM · 2022-11-03

**Confidence:** 3
**Correctness:** 3
**Technical Novelty And Significance:** 3
**Empirical Novelty And Significance:** 3
**Recommendation:** 8

**Clarity, Quality, Novelty And Reproducibility:**

This paper is well-written with providing clear formulation of unlearning properties and well motivated designs.

The method is technically sound, and some novel ideas are proposed through problem formulation, unlearning operator design and evaluation.

Codes are provided for reproducibility.


**Strength And Weaknesses:**

**Strength**
- The paper provides a clear formulation of required unlearning properties, which motivates the design of the optimization objective
- The proposed layer-wise deletion operator is efficient, flexible and provides theoretically bound for deleted edge prediction
- Extensive evaluation is conducted to showcase the performance of the proposed method

**Weaknesses**
- Performance on node deletion task seems to be less impressive

**Questions**
- The condition of $u\in S^l_{uv}$ in Eq. (3) is a bit hard to understand under the context. Can the authors provide more explanations about when this condition will be true and what part of the extended GNN layer will take $\phi$?

- What is the difference between $\mathcal{E}_t$ (which seems to be undefined and I guess it is the test set) and $\mathcal{E}_d$?


**Summary Of The Paper:**

This paper proposes an efficient graph unlearning method based on novel formalization of two unlearning properties, namely deleted edge consistency and neighborhood influence. A model-agnostic layer-wise deletion operator is optimized via an objective based on these two properties. Empirically results on link prediction and deleted edge prediction tasks demonstrates the unlearning effect.


**Summary Of The Review:**

This work introduces some new insights to the graph unlearning problem, and the presentation is easy to follow. I believe the community should find this work interesting, thus I lean to acceptance.

---

> ### Author Response · Authors · 2022-11-14
> **Response to Reviewer mYMM**
>
> We thank the reviewer for their insightful comments, and for acknowledging the significance of our contributions. Below, we address specific weaknesses and questions raised by the reviewer.
>
> **Performance on node deletion task seems to be less impressive**
>
> We would like to bring to the reviewers’ notice that in Section 5.2 of the manuscript we describe our results for node deletion tasks, where we delete $100$ nodes and their associated edges from the training data and evaluate the unlearning performance using the Membership Inference attack metric.
>
> In response to the reviewers’ feedback, we repeated our node deletion experiments on the DBLP dataset and done a complete hyperparameter search. These new results show that GNNDelete outperforms GraphEditor by 18% in accuracy and 19% in F1 score.  It is also 14% better than GraphEditor in terms of membership inference attacks
>
>
> |                                       | Acc          | F1           | MI Ratio     |
> |---------------------------------------|--------------|--------------|--------------|
> | Retrain from scratch (reference only) | 0.845 ±0.008 | 0.841 ±0.004 | 1.515 ±0.034 |
> | Gradient ascent                                  | 0.392 ±0.026 | 0.341 ±0.035 | 1.021 ±0.113 |
> | D2D                                                    | 0.250 ±0.000 | 0.250 ±0.000 | **1.755 ±0.065** |
> | GraphEraser                                       | 0.718 ±0.014 | 0.716 ±0.011 | 0.975 ±0.083 |
> | GraphEditor                                        | _0.765 ±0.012_ | _0.749 ±0.006_ | 1.260 ±0.088 |
> | GNNDelete (Ours)                              | **0.793 ±0.016** | **0.768 ±0.009** | _1.401 ±0.082_ |
>
>
>
>
>
> **More details about the condition u∈Suvl in Eq. (3)**
>
>
> We apologize for the lack of details and would like to clarify. When we want to delete an edge $e_{uv}$, $S^l_{uv}$ refers to the $l$-th hop local enclosing subgraph of $e_{uv}$, with $u$ being some other node in $S^l_{uv}$ (thus the notation is overloaded in Eq. 3, which we corrected). We will replace $u$ with some other letter, e.g., $w$ for clarifying that point such that it will be $w \in S^l_{uv}$.
>
> Further, the deletion operator is applied to each node embedding vector, acting as a mask, dependent on whether the node belongs to the specific enclosing subgraph or not. In particular, when a node is out of the enclosing subgraph, the deletion operator does not affect the node embedding at all. In contrast, when a node lies in the enclosing subgraph, function $\phi$ is applied to the node embedding vector. Thanks for bringing up this point, which we clarify it in the revised manuscript.
>
>
> **Difference between Et (which seems to be undefined and I guess it is the test set) and Ed?**
>
> We apologize for the confusion. $E_t$ represents the test edge set, which contains *existent* edges from the remaining edge set, in order to evaluate the performance of deletion. On the other hand, $E_d$ refers to the set of edges we ask the model to delete. We fixed the definition of $E_t$ in the revision in Section 5.2 and in a more detailed manner in Appendix C.1, along with an elaborate description of the performance metrics.
>
> **Conclusion**
>
> You can find the changes in the revised manuscript. We would like to thank you again for your valuable feedback, which made our paper stronger!

---

> ### Author Response · Authors · 2022-11-23
> **Looking forward to hearing from you**
>
> Dear reviewer,
>
> We sincerely appreciate your valuable comments on our work. In our previous response and the updated manuscript, we have tried our best to address the points raised in your review. Is there any unclear point that we can further clarify?
>
> Thank you again!

---

> ### Author Response · Authors · 2022-11-29
> **Looking forward to hearing from you**
>
> We sincerely appreciate your valuable comments on our work. In our previous response and the updated manuscript, we have tried our best to address the points raised in your review. Is there any unclear point that we can further clarify?
>
> Thank you again!

---

### Official Review · Reviewer_PAk1 · 2022-11-04

**Confidence:** 4
**Clarity, Quality, Novelty And Reproducibility:** As given in the list of weaknesses, t…
**Correctness:** 3
**Technical Novelty And Significance:** 3
**Empirical Novelty And Significance:** 3
**Recommendation:** 5

**Strength And Weaknesses:**

The strengths:
1) comparing to the partitioning based models, the proposed solution doesn't need to specify partitioning parameters that may have impact on the unlearning performance.
2) the proposed method was shown to have better performance than baselines in evaluation

The weaknesses
1) The layer-wise unlearning can be expensive. Although the number of trainable parameters to update in the unlearning process is independent of the size of the graph, the computational cost depends on the number of layers in graph neural networks.  The time complexity evaluation shows only on the 2-layer graph neural networks (GNNs). When the networks get deeper and have a larger number of parameters, the proposed model will become expensive.
2) The evaluation results were not well discussed. For example, \mathcal{E}_d is the set of edges to delete. what is  \mathcal{E}_t in Table 1 and 2?  If the target of GNNDELETE is to delete \mathcal{E}_d,  why improving the performance of GNNs on predicting the links in \mathcal{E}_d?
3) The unlearning impact should be also evaluated on other tasks. For example, if the request is to delete edges in \mathcal{E}_d, node classification performance should also be evaluated in the whole remaining graph, to see if the classification performance is retained for unaffected nodes.
4) when the unlearning task is about nodes (deleting nodes), how the  loss functions are defined?  the evaluation in Table 3 shows the performance of MI. What about the influence on link prediction?  if a set of nodes were deleted, how the link prediction is affected?
5) the evaluation mentioned node feature update unlearning tasks. However, there is no experiments about such tasks.

**Summary Of The Paper:**

This paper studies the problem of graph neural network unlearning. Unlike the partitioning-based unlearning strategies, the proposed solution works on two defined loss functions that reflect the target of edge deletion kind of unlearning request. One loss function measures the Deleted Edge Consistency: the predictiveness of deleted edges comparing to the nonexistent edges. The other loss function computes the Neighborhood Influence: the influence of edge deletion on local subgraphs. The unlearning process is to optimize these two loss functions over each layer, to reach the aim of unlearning about the edge deletion request.

**Summary Of The Review:**

The paper presents an interesting idea. However, there are several issues about evaluation results to address.

---

> ### Author Response · Authors · 2022-11-18
> **Response to Reviewer PAk1 [1/4]**
>
> Thank you for your helpful feedback! We appreciate that you felt our method is reasonable and that you recognize the results are impressive and valuable for many applications. Below, we briefly respond to each of your concerns. If, after reading our responses, you are still not convinced of the merits of our work, we would love to know what else we can do to help improve the work towards a better score in your eyes.
>
> **RE Q1**:
>
> Thank you for bringing up the computational cost of layer-wise operation. In fact, the real computational cost might be incurred by the neighborhood size of the target node/edge to be deleted + the number of deleted nodes/edges.
>
> We would like to make three points in response to your comment:
> 1. Many GNN models are rather shallow, as evidenced by several large-scale empirical studies, e.g., [1,6], which observed state-of-the-art performance across many small and large datasets is achieved by models with a handful number of layers (often 2-3 layers). Further, it has been established that overly deep GNN models can lead to over-smoothing and over-squashing problems [4-5].
> 2. Linear computational efficiency in the number of GNN layers is not necessarily a disadvantage. Further, Figure 2 in Section 5.2 shows that GNNDelete is faster than other graph unlearning methods. For example, it performs deletion 9.3 times faster than GraphEraser and 18.5% faster than GraphEditor.
> 3. To further analyze GNNDelete in response to your comment, we performed a new experiment applying GNNDelete only at the final/last GNN layer, finding that a simplified version of the GNNDelete can achieve comparable performance to the layer-wise version. Specifically, we conducted an experiment that compares layer-wise GNNDelete and the final-layer-only GNNDelete on the DBLP dataset. For the final layer-only version, we inserted the deletion operator after the final GNN layer only and calculated the loss in Eq. 7 at the final layer. The results below show that final-layer-only GNNDelete can achieve comparable results with layer-wise GNNDelete, with less than a 1% difference in link prediction performance and deletion performance (i.e., distinguishing deleted vs. non-deleted edges).
>
>
> **Comparison of layer-wise and final-layer-only GNNDelete. Deletion task: 2.5% edge deletion on DBLP. Evaluation: predictive performance (downstream link prediction) and deletion performance (distinguishing deleted vs. non-deleted edges).**
> |                                       | AUC on $E_t$           | AUC on $E_d$           |
> |---------------------------------------|--------------|--------------|
> | Retrain from scratch (reference only) | 0.964 ±0.003 | 0.506 ±0.013 |
> | GNNDelete - layer-wise (Ours)          | 0.934 ±0.002 | 0.748 ±0.006 |
> | GNNDelete - final layer (Ours)        | 0.927 ±0.005 | 0.739 ±0.005 |
>
> Related works:
> [1] Auto-GNN: Neural Architecture Search of Graph Neural Networks, arXiv:1909.03184
> [2] Deep Graph Neural Networks with Shallow Subgraph Samplers, ICLR 2021
> [3] Semi-Supervised Classification with Graph Convolutional Networks, ICLR 2017
> [4] Deeper insights into graph convolutional networks for semi-supervised learning. AAAI 2018
> [5] Revisiting graph neural networks: All we have is low-pass filters. arXiv:1905.09550
> [6] Design Space for Graph Neural Networks, NeurIPS 2020
>
> *Continuation of the response in the thread.*

---

> > ### Author Response · Authors · 2022-11-18
> > **Response to Reviewer PAk1 [2/4]**
> >
> > **RE Q2**:
> > Many thanks for raising this. Sorry for the confusion. $E_t$ represents the **test edge set** that is used to evaluate the predictive performance of the deleted GNN predictor. $E_d$ represents the **deleted edge set** used to evaluate deletion performance. To demonstrate the robustness of our method, we compare performance on both $E_t$ and $E_d$. We made it clear in our revision.
> >
> > We first describe the definition of evaluation metrics and then discuss the results.
> >
> > #### Definition of evaluation metrics (see updated Section 5.1 and Appendix C)
> > 1. **AUC on $E_t$: test model’s ability of link prediction (Shown in Table 1-2)**
> > This standard metric is the same as in other papers that evaluate the performance of GNN predictors (not deletion models).
> >
> > 2. **AUC on $E_d$: test a model’s ability to distinguish remaining edges (Er) from deleted edges (Ed) (Shown in Table 1-2)**
> > The idea is that the remaining edges (Er) should have high predicted probabilities, while the deleted edges (Ed) should have low predicted probabilities, i.e. g(Ed) < g(Er), where g is the link prediction decoder. When calculating AUC, we set the labels of g(Ed) to 0, since they do not exist after deletion. We set the labels of g(Er) to 1, since they exist. Therefore, the higher the AUC on Ed, the better the model can distinguish Ed from Er.
> >
> >
> > To calculate AUC on $E_d$, we prepare the prediction and ground truth vectors as follows
> > * Prediction: a concatenation of {$g(n_i, n_j) | e_{ij} \in E_d $} (edge probability of $E_d$) and {$g(n_k, n_l) | e_{kl} \in E_r $} (edge probability of randomly sample edges from E_r, with equal size as $E_d$)
> > * Ground truth: $|E_d|$ 0s and $|E_d|$ 1s
> >
> > These two vectors are fed into the AUC function to get the AUC score on $E_d$. We repeat the same process 500 times to sample different randomly sampled edges from $E_r$ and report the average score and standard deviation.
> >
> > 3. **MI Ratio: quantifies privacy leakage (shown in Table 3)**
> >
> > Let MI(.) denote the membership inference model
> >
> > a. Get the existence probability of Ed before deletion, MI_before($E_d$)
> > b. Get the existence probability of Ed after deletion, MI_after($E_d$)
> > c. Compute the before-to-after ratio of existence probability, MI ratio = MI_before($E_d$) / MI_after($E_d$)
> > If the ratio is higher than 1, it means that the model contains less information about $E_d$. Thus, it has forgotten $E_d$, which is the desired result. If the ratio is less than 1, it means the deleted model contains more information about $E_d$, which is not desired (quantified by the MI attack model).
> >
> > #### Discussion of results
> > We compare GNNDelete to several baselines on _edge deletion tasks_ and present the results in Tables 1-3. Table 1-2 shows the link prediction performance and ability to distinguish deleted edges from non-deleted edges. Table 3 shows the MI performance.
> >
> > Across three GNN architectures, we find that GNNDelete achieves the best performance on link prediction performances (AUC on $E_t$) (Table 1-2), outperforming GraphEditor and GraphErasor by 13.9% and 38.8%. Further, we observe that GNNDelete has the best performance at distinguishing deleted edges from non-deleted edges (AUC on $E_d$) (Table 1-2), outperforming GraphEditor and GraphEraser by 32.2% and 25.4%. GNNDelete even outperforms Retrain-From-Scratch by 21.7% under this setting, proving its capability of effectively unlearning the deleted edges. Interestingly, none of the existing baseline methods have comparable performance to GNNDelete on these performance metrics, including GraphEraser, which ignores the global connectivity pattern and overfit to specific shards, and GraphEditor, whose choice of linear architecture strongly limits the power of the GNN unlearning.  Our results demonstrate that baselines like Descent-to-Delete and Gradient Ascent lose almost their predictive prowess in making meaningful predictions and distinguishing deleted edges because the weight updates are independent of the unlearning task and affect all the nodes, including nodes associated with $E_t$. Please refer to the Appendix for results on PubMed (Tables 8-9), DBLP (Tables 10-11), OGB-Collab (Tables 12-13), and WordNet18 (Tables 14-15) using a deletion ratio of 0.5%, 2.5%, and 5%.
> >
> > Results in Table 3 show the MI performance on the DBLP and Wordnet18 on edge deletion tasks. It shows that GNNDelete outperforms baselines for most GNN models, highlighting its effective privacy protection. Across five GNN architectures, we find that GNNDelete improves on the MI ratio score of all baselines: GraphEditor (+9.7%), GraphEraser (+22.5%), Retrain-From-Scratch (+8.6%), Gradient Ascent (+23.5%), and Descent-to-Delete (+2.4%).
> >
> > *Continuation of the response in the thread.*

---

> > > ### Author Response · Authors · 2022-11-18
> > > **Response to Reviewer PAk1 [3/4]**
> > >
> > > **RE Q3**:
> > > Thank you for your great comment on evaluating the unlearning impact on other downstream tasks when deleting edges, such as node classification.
> > >
> > > In response to your comment, we considered three downstream prediction tasks: 1) node classification, 2) link prediction, and 3) graph classification. We performed 2.5% edge deletion using the unlearning method on the DBLP dataset and evaluated the deleted GNN predictor on all three downstream tasks and across three different GNN architectures. This means that, for example, if the request is to delete edges in $\mathcal{E}_d$, node classification or link prediction or graph classification performance is also evaluated on the whole remaining graph to see if the classification performance is retained for unaffected nodes.
> > >
> > > **Deletion task: 2.5% edge deletion on DBLP. Evaluated downstream task: node classification. Metric: Accuracy (higher is better)**
> > > |                                       |    Acc on GCN     |    Acc on GAT     |    Acc on GIN     |
> > > |---------------------------------------|:-----------:|:-----------:|:-----------:|
> > > | Retrain from scratch (reference only) | 0.821±0.012 | 0.830±0.012 | 0.817±0.012 |
> > > | Gradient ascent                       | 0.532±0.055 | 0.547±0.024 | 0.550±0.041 |
> > > | D2D                                   | 0.519±0.008 | 0.511±0.004 | 0.513±0.007 |
> > > | GraphEraser                           | 0.681±0.037 | 0.694±0.022 | 0.673±0.033 |
> > > | GraphEditor                           | 0.667±0.024 | 0.673±0.019 | 0.680±0.031 |
> > > | Ours                                  | 0.802±0.018 | 0.799±0.015 | 0.812±0.017 |
> > >
> > > **Deletion task: 2.5% edge deletion on DBLP. Evaluated downstream task: link prediction. Metric: AUC (higher is better)**
> > > |                                       | AUC on GCN   | AUC on GAT   | AUC GIN      |
> > > |---------------------------------------|--------------|--------------|--------------|
> > > | Retrain from scratch (reference only) | 0.964 ±0.003 | 0.956 ±0.002 | 0.931 ±0.005 |
> > > | Gradient ascent                       | 0.555 ±0.066 | 0.501 ±0.020 | 0.700 ±0.025 |
> > > | D2D                                   | 0.555 ±0.066 | 0.500 ±0.000 | 0.500 ±0.000 |
> > > | GraphEraser                           | 0.527 ±0.002 | 0.538 ±0.013 | 0.517 ±0.009 |
> > > | GraphEditor                           | 0.776 ±0.025 | 0.776 ±0.025 | 0.776 ±0.025 |
> > > | GNNDelete (Ours)                      | 0.934 ±0.002 | 0.914 ±0.007 | 0.897 ±0.006 |
> > >
> > > **Deletion task: 2.5% edge deletion on DBLP. Evaluated downstream task: graph classification. Metric: AUC (higher is better)**
> > > |                                       | AUC on GCN   | AUC on GAT   | AUC on GAT   |
> > > |---------------------------------------|--------------|--------------|--------------|
> > > | Retrain from scratch (reference only) | 0.758 ±0.068 | 0.763 ±0.049 | 0.769 ±0.047 |
> > > | Gradient ascent                       | 0.601 ±0.063 | 0.612 ±0.053 | 0.613 ±0.057 |
> > > | D2D                                   | 0.572 ±0.013 | 0.546 ±0.010 | 0.591 ±0.021 |
> > > | GraphEraser                           | 0.596 ±0.032 | 0.611 ±0.029 | 0.607 ±0.027 |
> > > | GraphEditor                           | 0.613 ±0.035 | 0.624 ±0.028 | 0.620 ±0.027 |
> > > | GNNDelete (Ours)                      | 0.710 ±0.041 | 0.723 ±0.038 | 0.733 ±0.030 |
> > >
> > > Empirical experiments demonstrate that GNNDelete can effectively retain the performance of node classification, link prediction, and graph classification while accommodating deletion requests. On average, across three GNN architectures, GNNDelete outperforms GraphEditor (the strongest baseline) by 13% on node classification, 14% on link prediction, and 10% on graph classification. We provide results and experimental details in Section 5.2 and Appendix C in our latest revision.
> > >
> > >
> > > **RE Q4**: When extending to delete nodes with GNNDelete, the loss function is not changed because we only need to consider all edges affected by the deleted node, which is shown below:
> > >
> > > $W_D^{l^{*}}= \arg \min_{W_D^{l}} \mathcal{L}^l = \arg \min_{W_D^{l}}\lambda \mathcal{L}^l_{\text{DEC}} + (1 - \lambda) \mathcal{L}^l_{\text{NI}}$,
> > >
> > > where $\lambda \in [0, 1]$ is a regularization coefficient that balances the trade-off between the two properties.
> > >
> > > *Continuation of the response in the thread.*

---

> > > > ### Author Response · Authors · 2022-11-18
> > > > **Response to Reviewer PAk1 [4/4]**
> > > >
> > > > **RE Q5**:
> > > > Thank you for your valuable suggestion. We apologize for the confusion and answer your questions in the following.
> > > >
> > > > MI attack is independent of unlearning. We leverage existing MI models to quantify whether unlearning methods can ensure the privacy of deleted nodes/edges. In the context of graph unlearning, we consider two setups using the MI attackers to evaluate deletion performance:
> > > > 1. First, delete edge(s) and then use the MI attacker to check whether the deleted edges can still be inferred.
> > > > 2. First, delete node(s) and then use the MI attacker to check whether the deleted nodes can still be inferred.
> > > >
> > > > To answer your question about **Table 3 and the influence of deletion on link prediction**, Tables 1-3 show the results when we delete edges. Out of the 3 tables, Tables 1 and 2 show the influence of edge deletion on link prediction performances, while Table 3 shows the performance of using an MI attacker to infer deleted edges. We apologize for the confusion.
> > > >
> > > > To answer the question of **how link prediction is affected if a set of nodes are deleted**, we delete 100 nodes on the DBLP dataset and evaluate the performance of the deleted GNN model for link prediction. The following results are also updated to the revised version in Table 6 in Appendix C.
> > > >
> > > > We can see that GNNDelete outperforms all baseline models on link prediction performance, including state-of-the-art graph unlearning models. GNNDelete is 24% better than GraphEditor and 42% better than GraphEraser in terms of AUC on the original test set for link prediction.
> > > >
> > > > **Deletion task: 100 node deletion. Evaluated downstream task: link prediction. Metric: AUC.**
> > > > |                                       | AUC on Et           |
> > > > |---------------------------------------|--------------|
> > > > | Retrain from scratch (reference only) | 0.973 ±0.002 |
> > > > | Gradient ascent                       | 0.571 ±0.032 |
> > > > | D2D                                   | 0.507 ±0.002 |
> > > > | GraphEraser                           | 0.513 ±0.004 |
> > > > | GraphEditor                           | 0.697 ±0.031 |
> > > > | GNNDelete (Ours)                      | 0.938 ±0.004 |
> > > >
> > > > **RE Q6**:
> > > > Thank you for your kind comment on the question of unlearning node features. Unlike existing graph unlearning methods, GNNDelete can be used to delete node features. To demonstrate this capability, we performed the following experiment where the unlearning approach is asked to delete 2.5% of node features on the DBLP dataset and the underlying GCN model. We evaluate the unlearned model across three downstream tasks.
> > > >
> > > > The results below show that GNNDelete can retain predictive performance on the remaining graph while deleting node features. It outperforms GraphEditor by 7% on downstream node classification, 17% on downstream link prediction, and 9% on downstream graph classification task.
> > > >
> > > >
> > > > **Deletion task: 2.5% node feature update on DBLP. Evaluated downstream tasks: node classification, link prediction, and graph classification.**
> > > > |                                       | Acc (node classification) | AUC (link prediction) | AUC (graph classification) |
> > > > |---------------------------------------|----------------------------|------------------------|-----------------------------|
> > > > | Retrain from scratch (reference only) | 0.810 ±0.021               | 0.897 ±0.025           | 0.753 ±0.042                |
> > > > | Gradient ascent                       | 0.614 ±0.042               | 0.655 ±0.030           | 0.623 ±0.046                |
> > > > | D2D                                   | 0.525 ±0.009               | 0.504 ±0.003           | 0.586 ±0.017                |
> > > > | GraphEraser                           | 0.682 ±0.044               | 0.579 ±0.021           | 0.592 ±0.032                |
> > > > | GraphEditor                           | 0.711 ±0.039               | 0.683 ±0.031           | 0.630 ±0.026                |
> > > > | GNNDelete (Ours)                      | 0.782 ±0.027               | 0.850 ±0.027           | 0.724 ±0.027
> > > >
> > > >
> > > >
> > > > **Conclusion**
> > > >
> > > > We thank again the reviewer for appreciating our contributions and asking great questions. These very detailed questions let us realize it is necessary to include more discussions and diverse tasks. We have incorporated answers to your questions and added experiments in our revision. We'd like to get in touch with you again to know if we have clarified your concerns and questions.

---

> ### Author Response · Authors · 2022-11-23
> **Looking forward to your feedback**
>
> Dear reviewer,
>
> Thank you again for your valuable suggestions on improving this manuscript. To address your concerns, we performed a number of additional experiments, included new tables and figures in the paper, and revised the text.
>
> We would love to know what you think about our response and if there is anything else we can do to improve the paper. We would greatly appreciate your considering increasing the score. Many thanks!

---

> ### Author Response · Authors · 2022-11-29
> **Looking forward to your response**
>
> Dear reviewer,
>
> Thank you again for your thoughtful commentary. Following your suggestions, we included new results and clarified notation in the initial submission. We would love to hear your thoughts on our response. Please let us know if there is anything else we can do to address your comments. We would be very grateful if you considered increasing the score.

---

> ### Author Response · Authors · 2022-12-07
> **Looking forward to hearing from you**
>
> Dear reviewer,
>
> We sincerely appreciate your valuable comments on our work. In our previous response and the updated manuscript, we have tried our best to address the points raised in your review. Is there any unclear point that we can further clarify?
>
> Thank you again!

---

> ### Author Response · Authors · 2022-12-13
> **Looking forward to your feedback**
>
> Dear reviewer,
>
> Thank you again for your valuable suggestions on improving this manuscript. Following your suggestions, we included a number of additional experiments, clarifications of formulations, and theoretical analysis.
>
> We would love to know what you think about our response and if there is anything else we can do to improve the paper. We would greatly appreciate your considering increasing the score. Many thanks!

---

### Author Response · Authors · 2022-11-18
**General Response**

We thank all reviewers for their time and valuable comments. We are happy that the reviewers find our method technically sound and interesting (Reviewers PAk1, myMM, kuQk, ctZT), note that the paper is well-written and well-motivated (Reviewers PAk1, myMM, kuQk, W6kL, ctZT), and mention that the experiments show broad success of GNNDelete (Reviewers PAk1, myMM, kuQk, W6kL, ctZT). In this general response, we clarify specific questions about our formulation, discuss the key additional experiments we conducted during the rebuttal, show the significance of our theoretical analysis, and also highlight the novelty of our work.


**Additional experiments show the utility of GNNDelete across deletion tasks, downstream prediction tasks, and multiple GNN predictors:**
1. In response to Reviewer kuQk’s comment, we performed additional experiments on *node classification* and *graph classification* downstream tasks. These new results in our response to individual reviewers and in the revised Appendix show a) that GNNDelete can be used to delete nodes, edges, and node features, b) that deleted GNN predictor can perform well across diverse downstream prediction tasks, and c) that GNNDelete can be integrated with a broad range of GNN predictors/architectures. Further, experiments show that GNNDelete outperforms baselines across these setups, with respect to the unlearned model performance. No other approach has all three capabilities in a-c).
2. Following the comments from Reviewers mYMM and ctZT, we repeated the node deletion experiments on the DBLP dataset through a complete hyperparameter tuning and showed that GNNDelete is able to outperform GraphEditor even in the case of node deletion tasks.
3. In response to Reviewer PAk1's comment, we performed a node feature update task, where GNNDelete shows a superior performance.
4. In response to Reviewer ctZT's comment, we, also, compared GNNDelete's performance with a dynamic network embedding method, and we showed that GNNDelete outperformed the latter.

**Formulation clarifications:**
1. We fixed the definitions of edge sets $E_d$ and $E_t$.
2. We rephrased the Problem Formulation to outline the graph unlearning problem.
3. We clarified the notation in Equation 3 on the GNNDelete’s deletion operator and the Equations regarding the loss functions.

**Theoretical analyses:**
1. We clarified that Theorem 1 gives an insight on how much node representations are going to be affected by the deletion operator. In addition, we point out that DEL is a layer-wise operator and as such, it does not drastically change node representations. Finally, we claim that GNNDelete is a safe-to-use operator, which can be integrated into state-of-the-art GNN models.
2. The Deleted Edge Consistency property states that for the case of a deleted edge $e_{ij}$, the node pair $(i,j)$ should have a representation that is similar to any random choice of a node pair $(k,l)$ in the graph (either that node pair is connected, meaning that $e_{kl}$ exists, or not, meaning that $e_{kl} does not exist).

**Novelty of our work:**
We appreciate that 4 out of 5 reviewers have explicitly acknowledged that contributions of this work are significant and that the approach is novel. In summary, present study makes the following key contributions:
1. We formalize machine unlearning on graphs through two properties (Deleted Edge Consistency and Neighborhood Influence) for which we empirically show that are both necessary for successful deletion.
2. We develop a GNN data deletion approach GNNDelete. It can unlearn node, edges, and node features from a trained GNN model, and can do so much more efficiently than existing graph unlearning methods without sacrificing predictive power.
3. GNNDelete is model-agnostic. It can be integrated with a broad range of GNN models, which we demonstrate in the paper.
4. Theoretically, we show the difference between node representations from the baseline and GNNDelete is bounded, which ensures GNNDelete’s strong performance.
5. Empirical results show that GNNDelete stands out among other graph unlearning methods by a) supporting deletion of edges, nodes, and node features and by achieving state-of-the-art performance on various downstream prediction tasks after the deletion.

Thanks for your efforts and valuable time on our manuscript. We address all concerns raised by reviewers and have updated the new version of the manuscript corresponding to your concerns and questions. The revised contents are highlighted in blue for ease of reading. If you have any questions or concerns, please feel free to let us know.

Best regards,
Authors of Paper2080

---

### Author Response · Authors · 2022-12-06
**Looking forward to your feedback**

Dear reviewers,

Thank you again for your precious time and valuable comments! Following your suggestions, we included new results, clarifications of formulations, and theoretical analysis. We would love to hear your thoughts on our responses. Please let us know if there is anything else we can do to address your comments. We would be very grateful if you considered increasing the score.

Thank you very much!

Best regards,
Authors of Paper2080

---

### Decision · Program_Chairs · 2023-01-20

**Decision:**

Accept: poster

**Justification For Why Not Higher Score:**

Some remaining minor weaknesses in clarity and experimental results.

**Justification For Why Not Lower Score:**

Novel problem, idea, approach.

**Metareview: Summary, Strengths And Weaknesses:**

The authors propose the graph unlearning problem. When GNNs are used on networks, there might be reasons for removing some information such as individual nodes but also edges between nodes (e.g., connections in a social network). The problem is now how we can unlearn the parts of the GNN that were influenced by these deleted elements during training, without having to retrain from scratch. The problem is novel and the proposed approach to achieving the unlearning objective is effective.

The reviewers provided helpful feedback, especially for the experimental evaluation. The authors responded to these suggestions and provided additional results.

One of the reviewers was very negative and gave a score of 3. However, even after several attempts by the authors and the AC, there was no activity of said reviewer during the discussion phase. In addition, the AC believes that the criticism of said reviewer was vague and unspecific. Hence, this reviewer's score was not taken into account in the recommendation.

**Note From Pc:**

if the above contains the word "oral" or "spotlight" please see: "oral" presentation means -> notable-top-5% and "spotlight" means -> notable-top-25%. As stated in our emails, we are disassociating presentation type from AC recommendations